# Unlearning with Asymmetric Sources: Improved Unlearning-Utility Trade-off with Public Data

## Abstract

Achieving certified data erasure in machine unlearning faces a fundamental trade-off: preserving model utility requires less noise, but formal privacy guarantees demand more. This tension typically degrades model performance. In this work, we study this challenge in Langevin Unlearning, a noisy variant of SGD that is uniquely amenable to theoretical analysis. We introduce an asymmetric unlearning setting assuming that datasets contain both private data (subject to unlearning) and public data (permanently retained). Our framework demonstrates that incorporating public data enables better unlearning-utility trade-offs without additional noise or restrictive differential privacy assumptions. We prove that public data volume quadratically reduces the Rényi divergence between unlearning and retraining distributions, allowing control over unlearning guarantees through data composition rather than noise amplification. The framework also provides a fine-grained analysis of how distributional alignment between public and private data affects performance preservation. Empirical validation using variational Rényi divergence estimation confirms our theoretical predictions, showing that strategic public data injection achieves comparable unlearning efficacy while significantly improving model performance and computational efficiency.

## 1 Introduction

The widespread adoption of machine learning across diverse applications has prompted regulatory responses aimed at protecting user privacy and data rights. Legislative frameworks such as the European Union's AI Act (Parliament & of the European Union, 2024) and Canada's Artificial Intelligence and Data Act (AIDA) (Parliament of Canada, 2022) establish fundamental principles including the "right to be forgotten", which mandates that individuals can request removal of their personal data from trained systems. This requirement presents significant technical challenges for modern machine learning paradigms, particularly deep learning and generative AI models that depend on large-scale datasets collected from public sources, often without explicit individual consent. Compounding this challenge, recent research demonstrates that neural networks exhibit a propensity to memorize training examples while maintaining generalization performance (Attias et al., 2024; Carlini et al., 2022; Nasr et al., 2023; Zhang et al., 2016).

The most straightforward approach to addressing data removal requests would be to retrain models from scratch after excluding the specified data points. However, this naive solution becomes prohibitively expensive for contemporary large-scale models, where training can require substantial computational resources. Moreover, the frequency of such requests in production systems would render this approach operationally impractical. This reality necessitates the development of machine unlearning techniques that can selectively remove specific data points from trained models while preserving overall performance. For certain applications, such removal should be certifiable through formal guarantees, ensuring that the unlearned model is statistically indistinguishable from one that was never trained on the removed data. Thus, effective unlearning algorithms must satisfy three fundamental requirements: provable erasure of target data, preservation of model utility, and computational efficiency that outperforms full retraining.

Most existing machine unlearning approaches operate under the assumption that *any* data point in the training set may require removal. While this assumption holds when working exclusively with sensitive datasets, it proves overly restrictive for real-world scenarios. Modern data collection pipelines aggregate information from heterogeneous sources, combining both sensitive private data and publicly available content. CommonCrawl (Common Crawl Foundation, 2024) and ImageNet (Deng et al., 2009) are examples of publicly available data used to train large language models and vision models. To our knowledge, the only prior work exploring mixed-privacy unlearning is Golatkar et al. (2021), who introduced Mixed-Linear Forgetting for computer vision tasks. Their approach requires architectural modifications to achieve forgetting through network linearization, limiting its general applicability. In the privacy-preserving machine learning literature, several works have shown that having access to a set of public data points allows for the design of algorithms with better privacy guarantees for the same amount of noise introduced into the model. When the public data distribution is close enough to the sensitive data distribution, these public data-assisted algorithms often offer a better privacy-utility trade-off than their conventional counterparts (Alon et al., 2019b; Amid et al., 2022; Ganesh et al., 2023a; Lowy et al., 2024).

In this work, we study the effect of considering that a portion of the training dataset is public and never subject to unlearning. We study this setting under Langevin Unlearning (Chien et al., 2024a), showing that restricting unlearning to private data improves guarantees. We ask the questions: *(1) Does adding public data improve Langevin Unlearning performance? (2) How does public-private distribution mismatch affect post-unlearning performance?* Our theoretical analysis provides clear answers. We first prove that injecting public data creates a more favorable initialization for the unlearning process (Theorems 3.1 and 3.2). We then provide a fine-grained analysis of the unlearning-utility trade-off, with our main contribution stated in Theorem 3.3, explaining how the distributional alignment between public and private data impacts the model's final performance. Finally, building on a variational representation of Rényi divergence (Birrell et al., 2023), we develop in Section 4.1 a framework for numerical evaluation of our bounds, showing that they capture some of the key dynamics of private-public learning and unlearning in practical settings.

## 2 BACKGROUND AND NOTATION

### 2.1 MACHINE UNLEARNING

Machine unlearning algorithms eliminate the influence of designated training data (the *forget set*) while balancing unlearning efficacy, model utility, and computational efficiency. Three canonical strategies illustrate the trade-offs: random re-initialization achieves perfect unlearning but destroys utility; retraining from scratch provides optimal guarantees but incurs prohibitive costs; no intervention preserves utility but achieves no unlearning. This motivates two paradigms: **Exact unlearning** replicates the retraining baseline through specialized architectures like SISA (Bourtoule et al., 2020) or Arcane (Yan et al., 2022), which enable targeted retraining but increase complexity. **Approximate unlearning** tolerates bounded discrepancies from retraining for practicality, including Newton-step updates (Golatkar et al., 2020) and noisy fine-tuning schemes like Langevin Unlearning (Chien et al., 2024a;b).

### 2.2 NOTATION

We consider probability distributions defined over a compact parameter space $\Theta$, where stochasticity arises from three sources: the weight initialization distribution $\pi_0$, the training data distribution $P_{\text{train}}$, and the inherent randomness of the optimization procedure. We denote by $\mathcal{P}(\Theta)$ the set of probability distributions supported on $\Theta$. Our analysis focuses on three parameter distributions: $\pi_L^T$ (the learning distribution after $T$ iterations of training on the full dataset), $\pi_U^K$ (the unlearning distribution after $K$ iterations of the unlearning procedure), and $\pi_R^T$ (the retraining distribution after $T$ iterations of training only on the retain set). A key quantity in our analysis is the Rényi divergence of order $\alpha$ between distributions $P$ and $Q$, denoted $D_\alpha(P\|Q)$, which we define rigorously in subsequent sections. We use $P_{\text{pub}}$ and $P_{\text{priv}}$ to represent the distributions of public and private data, respectively.

## 2.3 LANGEVIN UNLEARNING

A common approach to machine unlearning is to run a noisy projected gradient method starting from the trained weights, targeting a distribution close to retraining. Formally, at iteration $t$,

$$\theta_{t+1} = \Pi_\Theta[\theta_t - \eta\nabla_\theta\mathcal{L}(\theta_t) + \xi_t], \tag{1}$$

where $\mathcal{L}$ is a surrogate loss (e.g., empirical loss on a retain set), $\eta$ is the step size, and $\xi_t$ is injected noise (often Gaussian) controlling distributional closeness.

Langevin Unlearning (LU) (Chien et al., 2024a) instantiates this scheme with $\mathcal{L} = \mathcal{L}_{\mathcal{D}_r}$, the loss on the retain set, and $\xi_t \sim \mathcal{N}(0, 2\eta\sigma^2 I_d)$. This reduces to projected noisy gradient descent (PNGD) (pseudocode in Appendix A.5):

$$\theta_{t+1} = \Pi_\Theta\left[\theta_t - \eta\nabla_\theta\mathcal{L}_{\mathcal{D}_r}(\theta_t) + \sqrt{2\eta\sigma^2}\,W_t\right], W_t \sim \mathcal{N}(0, I_d) \tag{2}$$

LU provides certifiable approximate unlearning guarantees by minimizing the Rényi divergence between post-unlearning and post-retraining weight distributions (Chien et al., 2024a;b). However, these guarantees require that the *entire original training process* satisfies differential privacy (DP), necessitating PNGD with substantial noise injection from initialization. This requirement limits practical applicability, as it degrades model performance both before and after unlearning. In this work, we improve upon Chien et al. by relaxing the global DP assumption. Rather than requiring the entire learning process to satisfy DP, we assume only that the initialization distribution satisfies a log-Sobolev inequality—a mild condition naturally satisfied by standard Gaussian initialization. This property is preserved through PNGD iterations by Lemma A.1 due to loss smoothness. This relaxation enables us to derive data-dependent bounds that quantify how public data abundance improves unlearning without noise amplification, a key contribution unavailable in prior work. Concurrent approaches like (Koloskova et al., 2025) require only smoothness assumptions, but such data-agnostic bounds depend primarily on projection set geometry rather than training data structure.

## 3 ASYMMETRIC LANGEVIN UNLEARNING

**Motivation.** Our approach is motivated by a realistic data setting, well-established in the privacy-preserving machine learning literature, that leverages public data to improve the privacy-utility trade-off (Alon et al., 2019a; Ganesh et al., 2023b; Lowy et al., 2024; Amid et al., 2022). We introduce this asymmetric data model to Langevin Unlearning, which allows us to relax the restrictive Differential Privacy (DP) assumption over the entire dataset. By explicitly modeling this asymmetry, we can leverage public data to enhance the unlearning process to improve both efficacy and model performance without compromising privacy guarantees.

**Problem Setting.** We consider empirical risk minimization over a dataset $D = D_{\text{pub}} \cup D_{\text{priv}}$ comprising two components: a public set $D_{\text{pub}}$ with $n_{\text{pub}}$ samples from a distribution $P_{\text{pub}}$, and a private set $D_{\text{priv}}$ with $n_{\text{priv}}$ samples from a distribution $P_{\text{priv}}$. The training loss is $\mathcal{L}_D(\theta) = \frac{1}{n_{\text{pub}}+n_{\text{priv}}}\sum_{x\in D} l(\theta, x)$. Only the private data is subject to unlearning requests, while public data remains permanently available. We employ $T$ PNGD iterations with projections onto $\Theta \subset \mathbb{R}^d$ (radius $R$) to obtain $\theta_T$. Since PNGD injects Gaussian noise at each step, it induces probability distributions over parameters rather than deterministic iterates. In order to ensure that everything is well-behaved, one has to impose a regularity assumption on the initialization probability distribution.

**Definition 3.1.** *(Log-Sobolev inequality (Gross, 1975)) A probability measure $P \in \mathcal{P}(\mathbb{R}^d)$ satisfies a Log-Sobolev inequality with constant $C$ if*

$$\forall Q \in \mathcal{P}(\mathbb{R}^d), D_{KL}(Q\|P) \leq \frac{C}{2}I(Q, P), \tag{3}$$

*where $D_{KL}$ denotes the KL divergence and $I(Q,P) = E_Q\left[\|\nabla\log\frac{q}{p}\|^2\right]$ is the relative Fisher information.*

Our analysis compares two such distributions: the *unlearning distribution* $\pi_U$ (obtained by applying LU on the retain set from the trained model), and the *retraining distribution* $\pi_R$ (obtained by training

from scratch on the retain set). Following Chien et al. (2024a), we measure unlearning quality via Rényi divergence.

**Definition 3.2.** *For probability measures $P, Q$ with $P \ll Q$, their Rényi divergence of order $\alpha \in (0, +\infty) \setminus \{1\}$ is*

$$D_\alpha(P \| Q) = \frac{1}{\alpha - 1} \log \mathbb{E}_Q \left[ \left( \frac{dP}{dQ} \right)^\alpha \right],$$

*where $\frac{dP}{dQ}$ is the Radon-Nikodym derivative. This generalizes KL divergence ($\alpha \to 1$), reverse-KL ($\alpha \to 0$), and connects to $\varepsilon$-differential privacy in the limit $\alpha \to \infty$ (Mironov, 2017).*

**Our Contribution.** Our main contribution is showing that incorporating public data improves the unlearning-utility trade-off. While prior work proved that Langevin Unlearning's efficacy increases with noise magnitude (Chien et al., 2024a;b), this approach often degrades model performance. We break this dependency by introducing a new lever: the volume of public data. We demonstrate that increasing the amount of public data improves unlearning guarantees, i.e., lowers the Rényi divergence $D_\alpha(\pi_U \| \pi_R)$, without requiring additional noise injection or a global DP assumption. This allows for a fine-grained control over unlearning by adjusting data composition rather than simply amplifying noise.

### 3.1 DEFINING THE WEIGHT DISTRIBUTIONS

Consider the PNGD learning algorithm $\mathcal{A}$ applied to dataset $D = D_{\text{pub}} \cup D_{\text{priv}}$, where an unlearning request targets a subset $D_{\text{forget}} \subseteq D_{\text{priv}}$. Our analysis describes the relationship between three weight distributions arising from different training scenarios:

**Learning distribution $\pi_L^T$:** The weight distribution after $T$ PNGD iterations on the complete dataset $D$, starting from $\theta_0 \sim \pi_0$, a sample from the initialization distribution $\pi_0$. This represents the original trained model before any unlearning requests.

**Unlearning distribution $\pi_U^K$:** The weight distribution after $K$ PNGD iterations on the retain set $D \setminus D_{\text{forget}}$, initialized from $\theta_0 \sim \pi_L^T$. This captures the model state after applying our unlearning procedure to the originally trained model.

**Retraining distribution $\pi_R^T$:** The weight distribution after $T$ PNGD iterations on the retain set $D \setminus D_{\text{forget}}$, starting from the original initialization $\theta_0 \sim \pi_0$.

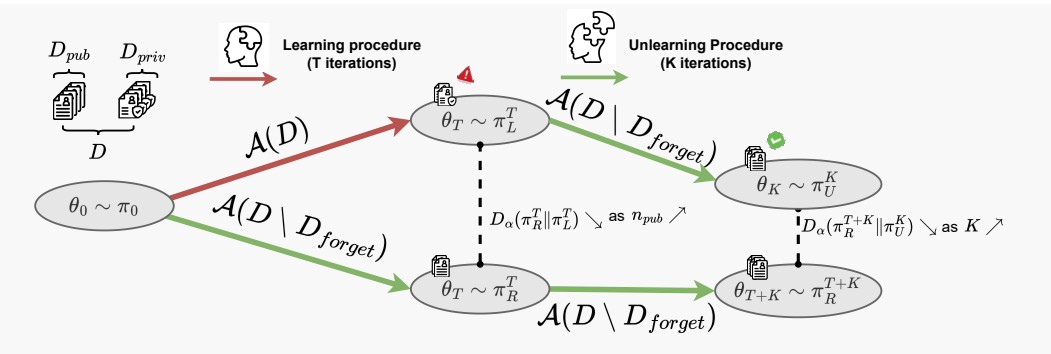

Figure 1: Training pipelines showing the relationship between learning, unlearning, and retraining with public data injection. The divergence $D_\alpha(\pi_R^T \| \pi_L^T)$ quantifies how public data helps maintain similarity between retraining and original learning distributions, facilitating subsequent unlearning.

The effectiveness of unlearning is measured by $D_\alpha(\pi_U^K \| \pi_R^{T+K})$, while the presence of public data helps control $D_\alpha(\pi_R^T \| \pi_L^T)$, creating favorable conditions for the unlearning process.

## 3.2 Unlearning Performance

We now present theoretical guarantees for asymmetric Langevin unlearning that demonstrate how public data fundamentally improves unlearning efficiency. Our analysis adapts the prior work of Chien et al. (2024a) by removing restrictive differential privacy assumptions, and providing explicit characterization of how public and private data contributions differ in the unlearning bounds. We also provide minor corrections to the bounds presented in Chien et al. (2024a); note, however, that these corrections do not change the key contributions and messages in (Chien et al., 2024a).

The following result explains how public data reduces reliance on differential privacy constraints, decoupling unlearning efficacy from model performance and enabling fine-grained analysis of this trade-off across different public-private distribution regimes (Section 3.3).

**Theorem 3.1** (The role of public data in shrinking the learning / retraining mismatch. )**.** *Suppose that the loss is $L$-smooth and $M$-Lipschitz, and that the initialization distribution satsifies a $C_0$-log Sobolev inequality. Moreover, suppose that the PNGD updates project onto a compact set $\Theta$ of radius $R$.*
*Then at learning iteration $T$, we have the following upper bound on the Renyi divergence between the retraining $\pi_R^T$ and learning $\pi_L^T$ distributions:*

$$\frac{D_\alpha(\pi_R^T \| \pi_L^T)}{\alpha} \leq \frac{2M^2\eta^2 n_{\text{forget}}^2}{(n_{\text{pub}} + n_{\text{priv}})^2\sigma^2} \sum_{t=1}^{T-1} \prod_{t'=t}^{T-1} \left(1 + \frac{\eta\sigma^2}{C_{t',1}}\right)^{-1},$$

*where $0 < C_{t',1} \leq (1+\eta L)^{2K}C_0 + 2\eta\sigma^2\frac{(1+\eta L)^{2K}-1}{(1+\eta L)^2 -1}$ are log Sobolev constants of the distributions of the intermediate PNGD updates. Using the support's radius allows to loosely upper bound those constants (Chien et al., 2024a): $C_{t',1} \leq 6e^{\frac{4\tau}{\eta\sigma^2}}(4\tau^2 + \eta\sigma^2)$ with $\tau = R + \eta M$.*

*Proof sketch. The proof follows the analytical framework of Chien et al. (2024a, Theorem 3.3), adapted to leverage the presence of public data in the training set. By distinguishing between public and private data contributions in the gradient updates, we reduce the privacy erosion (Chourasia et al., 2021) of each PNGD update.*

This bound reveals that we can fix noise magnitude $\sigma$ to be arbitrarily small to preserve performance while controlling the divergence through public data volume. When $n_{\text{pub}} \gg n_{\text{forget}}$, the learning and retraining distributions remain close regardless of noise level, providing favorable initial conditions for unlearning (Fig. 2b). Geometrically, for any fixed forget set size, the retraining distribution stays within a divergence ball whose radius shrinks quadratically with the number of public points.

**Theorem 3.2** (Convergence guarantee of Langevin unlearning (Chien et al., 2024a, Theorem 3.2))**.** *Suppose that the loss is $L$-smooth and $M$-Lipschitz, and that the learning distribution of weights at time $T$ satisfies a $C$ log-Sobolev inequality. Then, the Rényi divergence between $\pi_U^K$ (the unlearning distribution after $K$ iterations) and the retraining distribution after $T + K$ iterations is upper bounded by*

$$D_\alpha(\pi_R^{T+K} \| \pi_U^K) \leq D_\alpha(\pi_L^T \| \pi_R^T) \min\left(\prod_{k=1}^{K}\left(1 + \frac{2t\sigma^2}{(1+\eta L)^2 C_{U,k}}\right)^{\frac{-1}{\alpha}}, \exp\left(-\frac{2K\sigma^2\eta}{\alpha\tilde{C}}\right)\right),$$

*where $0 < C_k \leq (1+\eta L)^{2K}C + 2\eta\sigma^2\frac{(1+\eta L)^{2K}-1}{(1+\eta L)^2 -1}$, and $\tilde{C} \leq 6\left(4\tau^2 + 2\eta\sigma^2\right)\exp\left(\frac{4\tau^2}{2\eta\sigma^2}\right)$.*

*Moreover, if the loss function is $m$-strongly convex and the initial log-Sobolev constant satisfies $C > \frac{\sigma^2}{m}$, we get the following exponential decay of the Rényi divergence with respect to the unlearning iteration:*

$$D_\alpha(\pi_R^{T+K} \| \pi_U^K) \leq D_\alpha(\pi_L^T \| \pi_R^T) \exp\left(-\frac{2K\sigma^2\eta}{C\alpha}\right).$$

This theorem establishes the convergence guarantee for Langevin unlearning by showing that the Rényi divergence between the unlearning and retraining distributions decreases exponentially with unlearning iterations $K$, with the convergence rate controlled by the initial divergence $D_\alpha(\pi_R^{T+K} \| \pi_U^K)$. When combined with Theorem 3.1, this reveals the mechanism by which public data improves unlearning: the quadratic reduction in initial divergence from public data injection translates directly into tighter convergence bounds.

### 3.3 Performance Without Noise: The Role of Distribution Alignment

LU faces a fundamental dilemma: increasing noise improves unlearning guarantees but degrades model performance. Our asymmetric approach breaks this trade-off by leveraging public data abundance rather than noise amplification. However, the effectiveness of this strategy depends on the relationship between public and private data distributions.

We now analyze when public data injection preserves performance, and when it introduces new challenges. Our results reveal that performance preservation is not automatic – it depends on the distributional alignment between public and private data. When these distributions are similar, public data acts as a performance stabilizer, allowing effective unlearning without quality degradation. Conversely, when distributions differ significantly, performance impacts emerge, though they remain more controlled than noise-based approaches.

We evaluate post-unlearning performance on the private data distribution *only*, reflecting realistic deployment scenarios where the primary concern is maintaining model quality on the sensitive data that remains after unlearning. Performance analysis on the full mixture of public and private distributions is provided in Appendix A.4.1 for completeness.

**Theorem 3.3.** *Assuming the data generating distributions share the same support, that the weight space $\Theta$ is compact and that the loss is $M$-Lipschitz wrt $\theta$, we have the following upper bound on the generalization error on the private data after performing $K$ iterations of unlearning, and initializing a weight $\theta_0$ from $\pi_L^T$:*

$$\mathbb{E}_{\theta \sim \pi_U^K}\left[\mathbb{E}_{x \sim P_{\text{priv}}}[\mathcal{L}(\theta, x)]\right] \leq \underbrace{\exp\left(\frac{n_{\text{pub}}}{n_{\text{pub}} + n_{\text{retain}}} D_\infty(P_{\text{priv}} \| P_{\text{pub}})\right) \mathbb{E}_{\theta \sim \pi_R^{T+K}}\left[\mathbb{E}_{d \sim P_{\text{train}}}[\mathcal{L}(\theta, d)]\right]}_{\textit{distribution mismatch penalty}}$$

$$+ M \times diam(\Theta) \times \underbrace{\sqrt{\frac{1}{2} D_\alpha(\pi_R^{T+K} \| \pi_U^K)}}_{\textit{unlearning approximation error}},$$

*where $D_\infty(P \| Q) = \log\left(\operatorname{ess\,sup}_{x \sim Q} \frac{p(x)}{q(x)}\right)$ is the infinite Rényi divergence (worst case regret (Erven & Harremoës, 2014)) and $P_{\text{train}}$ denotes the mixture of distributions $D_{\text{pub}}$ and $D_{\text{priv}}$ used for training the model.*

*Proof sketch. The proof uses the Kantorovitch-Rubinstein duality Theorem A.1 to bound the performance gap by the dual of the Wasserstein distance between $\pi_U^K$ and $\pi_L^{T+K}$, then relates this to Rényi divergence via standard inequalities leveraging the compactness of the weight space $\Theta$. For private data evaluation, importance weighting introduces a mismatch penalty controlled by the worst case regret, $D_\infty(P_{\text{priv}} \| P_{\text{pub}})$, weighted by the public data fraction.*

This proposition enables a fine-grained analysis of the unlearning-performance trade-off. In the regime where $n_{\text{pub}} \to \infty$ (optimal for unlearning efficacy):

1. **Aligned distributions** ($D_\infty(P_{\text{priv}} \| P_{\text{pub}}) \approx 0$): The distribution mismatch penalty vanishes, and the unlearned model's performance on unseen private data is guaranteed to be at least as good as the retrained model's performance on the training mixture. This represents the ideal scenario where public data injection preserves performance.

2. **Misaligned distributions** ($D_\infty(P_{\text{priv}} \| P_{\text{pub}}) \gg 0$): The exponential penalty term dominates, causing the upper bound to become vacuous. While this confirms that performance degradation will occur, the bound's looseness prevents us from quantifying the actual extent of this degradation. The true performance impact may be better than this worst-case guarantee suggests.

**Retraining performance bound** ($\mathbb{E}_{\theta \sim \pi_{\mathbf{R}}^{\mathbf{T}}}\left[\mathbb{E}_{\mathbf{x} \sim \mathbf{p}_{\text{train}}}[\mathcal{L}(\theta, \mathbf{x})]\right]$): The upper bound could be further improved to include the *optimal* distribution, i.e by linking $\mathbb{E}_{\theta \sim \pi_{\mathbf{R}}^{\mathbf{T}}}\left[\mathbb{E}_{\mathbf{x} \sim \mathbf{p}_{\text{train}}}[\mathcal{L}(\theta, \mathbf{x})]\right]$ to $\arg\min_{\pi \in \mathcal{P}(\mathbb{R}^{\mathbf{d}})} \mathbb{E}_{\theta \sim \pi}\left[\mathbb{E}_{\mathbf{x} \sim \mathbf{p}_{\text{train}}}[\mathcal{L}(\theta, \mathbf{x})]\right]$. However, standard generalization bounds for Langevin dynamics (Raginsky et al., 2017; Xu et al., 2018) do not directly apply to our setting due to the

projection operator $\Pi_\Theta$ in the PNGD updates. These classical results focus on unconstrained non-convex optimization, whereas our bounded domain introduces additional complexity. The most relevant analysis we are aware of is Lamperski (2020), who study generalization properties of projected Stochastic Gradient Langevin Dynamics, though their work considers the infinite-data regime.

## 4 EXPERIMENTS

Our theoretical analysis provides upper bounds on the Rényi divergence $D_\alpha(\pi_R^{T+K}\|\pi_U^K)$ that governs unlearning performance. However, these bounds involve iteration-dependent log-Sobolev constants that are difficult to estimate in practice, making it unclear how tight our theoretical guarantees actually are. To gain empirical insight into the behavior of this divergence, we estimate its value using samples from the weight distributions. To our knowledge, this is the first attempt to evaluate unlearning performance through direct estimation of the Rényi divergence between the parameter distributions—moving beyond output-based unlearning evaluations to directly examine the parameter distributions. Building on Birrell et al. (2021; 2023), we leverage the variational representation of the Rényi divergence for numerical estimation.

**Theorem 4.1.** *(Convex conjugate variational approximation of the Rényi divergence (Birrell et al., 2023)) Let P,Q two probability distributions supported on $\Omega$, such that $P \ll Q$, and let $\mathcal{M}_b$ be the space of bounded measurable functions on $\Omega$. Then, $\forall \alpha \in (0, +\infty) \setminus \{1\}$,*

$$\frac{D_\alpha(P\|Q)}{\alpha} = \sup_{g \in \mathcal{M}_b(\Omega), g<0} \int g\, dQ + \frac{1}{\alpha - 1} \int |g|^{\frac{\alpha-1}{\alpha}} dP + \alpha^{-1} \left(\log \alpha + 1\right). \quad (4)$$

This variational representation of Rényi divergence allows us to obtain estimates of $D_\alpha(\pi_R^{T+K}\|\pi_U^K)$ using trained models as samples – to our knowledge, the first such attempt in the unlearning literature. We emphasize that this is not intended as a practical evaluation methodology for machine unlearning, as it requires training numerous models to obtain sufficient samples for reliable estimation. Standard approaches like membership inference attacks (MIAs) (Shokri et al., 2017; Carlini et al., 2021; Hayes et al., 2024) remain more suitable for practical evaluation. Our goal is purely investigative: to understand how the Rényi divergence behaves empirically and assess whether our theoretical bounds, despite containing hard-to-estimate constants, provide meaningful guidance in realistic scenarios.

We present our findings in two parts: Sections 4.1 and 4.2 investigate the behaviour of the upper bounds provided respectively in Theorem 3.2 and Theorem 3.3, while Section 4.3 provides standard membership inference attack and utility evaluations to contextualize our approach within existing unlearning assessment practices.

### 4.1 EVALUATING THE RÉNYI DIVERGENCE

**Experimental Setup.** We evaluate our approach on a multi-class image classification task using two domains from the DomainNet dataset (Peng et al., 2019): Quickdraw (sketches) and Clipart (stylized images), each containing 24 classes. We select these visually distinct domains to investigate how public-private data alignment affects unlearning and performance (Fig. 4).

The experimental configuration treats Clipart images as private data (subject to unlearning) and Quickdraw images as public data (permanently retained). For a training set of size $n = n_{\text{pub}} + n_{\text{priv}}$, we train models using cross-entropy loss and PNGD updates. To obtain samples from the weight distributions $\pi_U^K$ and $\pi_R^T$, we train $N$ models in parallel: one set undergoes unlearning (fine-tuning on the retain set after initial training), while another set trains from scratch on the retain set only. This procedure yields $N$ weight samples from each distribution, enabling empirical estimation of $D_\alpha(\pi_U^K\|\pi_R^T)$ through the variational formulation (Theorem 4.1).

**Estimation Method.** We approximate the variational Rényi representation (Eq. (4)) using neural network discriminators to parameterize the function space $\mathcal{M}_b(\Omega)$. This approach follows established practices in divergence estimation (Birrell et al., 2021; 2023; Belghazi et al., 2021) (pseudo-code in Appendix A.7.3) . To reduce estimation variance, we apply spectral normalization (Miyato et al., 2018) to regularize the discriminator networks. Complete details on discriminator architecture and training procedures are provided in Appendix A.7. **Results.** Fig. 2a presents our Rényi

estimation results, demonstrating the effectiveness of public data injection for improving unlearning efficiency. The experiments are conducted using $N = 30,000$ models for each distribution and averaged across 5 discriminator trainings with spectral normalization. The PNGD noise scale is $\sigma = 0.01$ and $\alpha = 2$. The results show that increasing public data volume reduces $D_\alpha(\pi_R^{T+K} \| \pi_U^K)$, with the divergence decreasing both as a function of unlearning iterations and public data proportion. To understand the mechanism driving these improvements, we conduct an ablation study examining the initial conditions after a *single* unlearning iteration. Fig. 2b solates the effect of public data on the starting distributions by measuring $D_\alpha(\pi_R^{T+1} \| \pi_U^1)$ as a function of public data volume. Rather than directly improving the unlearning procedure itself, public data creates more favorable initial conditions by ensuring the learning and retraining weight distributions begin in closer proximity. This mechanistic understanding validates our theoretical framework: public data primarily controls the initial gap between distributions (Theorem 3.1), which then propagates through the unlearning iterations to produce the final performance gains. Table 1 reports test accuracy for unlearned and retrained models across different public/forget splits. Surprisingly, despite the public and private data distributions being markedly different, the two procedures yield nearly identical accuracy (differences $\leq 0.05$). This observation indicates that the excess-risk bound in Proposition 3.3 can be overly conservative. Hence, Langevin unlearning empirically achieves retraining-level generalization even under unfavorable distribution shifts for this task. Identifying the structural conditions under which this distributional term becomes negligible remains an important direction for future work.

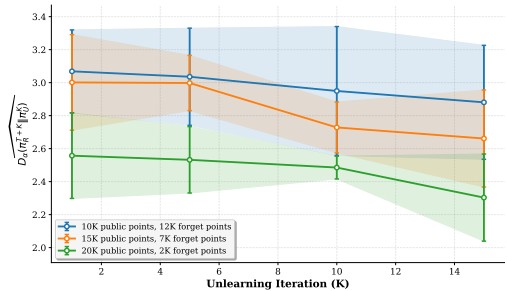
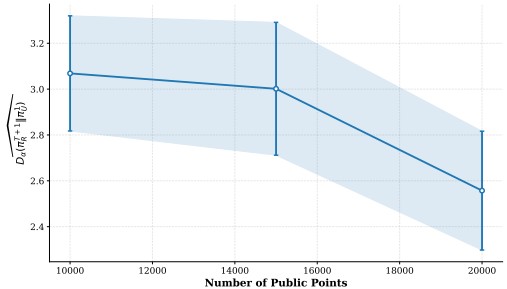

(a) Variational Rényi divergence estimation as a function of public data proportion in the training set. The results demonstrate that increasing public data volume reduces $D_\alpha(\pi_R^{T+K} \| \pi_U^K)$, confirming improved unlearning efficacy. This divergence also decreases with the unlearning iterations.

(b) Ablation study: Initial distribution alignment as a function of public data volume. The Rényi divergence $D_\alpha(\pi_R^{T+1} \| \pi_U^1)$ between retraining and unlearning distributions after a single unlearning iteration decreases as the number of public data points increases.

Figure 2: Rényi divergence estimation for a different number of clipart images (public set)

## 4.2 DISTRIBUTION ALIGNMENT AND THE UNLEARNING-UTILITY TRADE-OFF

Theorem 3.3 characterizes a trade-off caused by public data injection: as we increase public data volume, the *unlearning approximation error* decreases, yet the *distribution mismatch penalty* simultaneously grows. The balance between these competing terms determines whether public data injection preserves or degrades model performance. To empirically investigate this trade-off, we conduct experiments across two distinct distributional regimes: one where the public and private domains exhibit moderate visual alignment, and another where they are substantially misaligned.

We fix $K = 5$ unlearning iterations and evaluate performance using the DomainNet dataset across two domain pairs. The **aligned regime** pairs Quickdraw (public) and Clipart (private), which despite visual stylistic differences share semantic structure. The **misaligned regime** pairs Infograph (public) and Real (private), which exhibit greater distributional divergence. We measure model performance via loss on the private data distribution $P_{\text{priv}}$ after unlearning, comparing against the retraining baseline on the training mixture. Results are summarized in Table 1.

The results reveal a contrast between the two regimes. In the **aligned setting**, the relative performance gap remains modest (3.68–4.62%) across varying public data volumes, suggesting that the mismatch penalty remains manageable and the approximation error reduction dominates. In con-

Table 1: Unlearning vs Retraining Performance Across Distribution Alignments, $K = 5$

| Public Domain | Private Domain | Public Points | Private Points | Forget Set | Unlearn Avg. Loss | Retrain Avg. Loss | Rel. Diff (%) |
|---|---|---|---|---|---|---|---|
| Quickdraw | Clipart | 10000 | 20000 | 10000 | 3.102 | 2.976 | 4.23 |
| Quickdraw | Clipart | 30000 | 20000 | 10000 | 3.102 | 2.965 | 4.62 |
| Quickdraw | Clipart | 40000 | 20000 | 10000 | 3.099 | 2.989 | 3.68 |
| Infograph | Real | 10000 | 20000 | 10000 | 2.233 | 2.495 | 10.53 |
| Infograph | Real | 30000 | 20000 | 10000 | 2.238 | 2.496 | 10.34 |
| Infograph | Real | 40000 | 20000 | 10000 | 2.233 | 2.504 | 10.81 |

trast, the **misaligned setting** exhibits a persistent performance gap (10.34–10.81%), with minimal sensitivity to public data volume. This indicates that when distributional divergence is large, increasing public data fails to overcome the mismatch penalty, rendering the approximation error reduction insufficient to improve generalization.

### 4.3 PRACTICAL EVALUATION OF LU IN THE ASYMMETRIC SETTING

We now adopt standard evaluation methodology from the unlearning literature Hayes et al. (2024), introducing easily reproducible experiments which highlight that public data can benefit machine unlearning (LU). We provide an overview here and defer details to Appendix A.8.

**Evaluation Method.** This evaluation is based on the U-LiRA membership inference attack for unlearning (Hayes et al., 2024; Carlini et al., 2021). Given a training set, forget set, and specified learning and unlearning algorithms, the adversary's goal is to infer whether a model's weights $\theta$ were drawn from the unlearning distribution $\pi_U^K$ or the retraining distribution $\pi_R^{T+K}$. Intuitively, lower attack accuracy indicates that the unlearning and retraining distributions are harder to distinguish, i.e., better unlearning.

In its most basic form, U-LiRA can be formalized via Bayes' rule under a uniform prior on whether the forget set was included during training. Letting $P(\theta \mid \cdot)$ denote the likelihood of observing model parameters $\theta$ under a given distribution, and $P(\cdot \mid \theta)$ as the posterior probability that $\theta$ was drawn from that distribution, we have

$$P(\pi_U^K \mid \theta) = \frac{P(\theta \mid \pi_U^K)}{P(\theta|\pi_U^K) + P(\theta \mid \pi_R^{T+K})}.$$

By selecting a one-dimensional representation of the models $f : \Theta \to \mathbb{R}$ and assuming that the induced distributions $f_\sharp \pi_U^K$ and $f_\sharp \pi_R^{T+K}$ are Gaussian, we can estimate the likelihood terms $P(\theta \mid \cdot)$ from a tractable number of model samples.

**Experimental Setup.** For the sake of completeness, we focus this next set of experiments on a completely different task, namely sentiment analysis on the IMDB dataset of movie reviews (Maas et al., 2011). This is a simple binary classification task, where an LSTM (Hochreiter & Schmidhuber, 1997) learns to recognize if a review is either negative or positive. We use the Amazon reviews dataset from Zhang et al. (2015) as the public data source. We use a forget set of 100 uniformly sampled examples from the IMDB dataset. For both experiments, i.e., with and without public data injection, we generate $N = 50$ models to estimate each likelihood density, and report the empirical distribution of probabilities assigned to the right

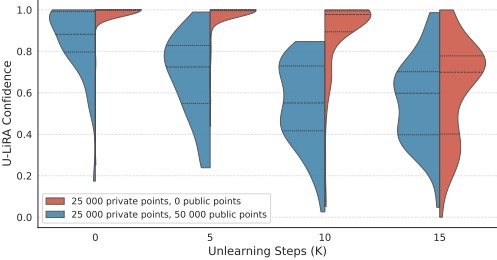

Figure 3: U-LiRA confidence scores after K unlearning iterations as violin plots with quartiles.

origin distribution by U-LiRA (confidence scores) for 50 models test (25 from $\pi_U^K$, and 25 from $\pi_R^{T+K}$, where $T = 50$ and $K = 1 \to 15$). Fig. 3 highlights that without public data injection, U-LiRA is able to identify a large proportion of models confidently and correctly, even after a number of unlearning steps. This observed discriminative power is heavily impacted by public data injec-

tion. We can also observe that modes of the confidence scores generally decrease with the number of unlearning steps, highlighting the unlearning effectiveness of LU.

Now that we've observed the effect of public data injection on the unlearning effectiveness of LU, we change our focus towards its impact on model utility. To this end, we report in Table 2 the average model accuracies over the 75 models we trained for each model distribution, on a test set of $10,000$ unseen samples from the IMDB dataset. As the Amazon reviews dataset appears to be a good auxiliary public data source for the IMDB review classification problem (close data distributions), we also include an experiment in which a uniformly sampled $40\%$ of its labels are flipped, thus increasing distribution mismatch between public and private sources.

Table 2: Unlearned and Retrained Model Test Accuracies for Different Scenarios

| Private Dataset | Private Points | Public Dataset | Public Points | Flipped Public Labels | Unlearned Accuracy (%) | Retrained Accuracy (%) |
|---|---|---|---|---|---|---|
| | | None | 0 | 0% | 82.59 | 82.54 |
| IMDB | 25,000 | Amazon Reviews | 50,000 | 0% | 81.42 | 82.15 |
| | | Amazon Reviews | 50,000 | 40% | 80.40 | 80.80 |

From Table 2, we can observe that model accuracy does decrease from the injection of public data. However, this drop in accuracy is rather negligible compared to the extent to which public data injection improves the unlearning effectiveness of LU, which is highlighted by Fig. 3. As expected, the drop in accuracy is proportionately much lower when the quality of auxiliary public data is high ($1.17\%$ for unlearned and $0.39\%$ for retrained) than when it is low ($2.19\%$ for unlearned, an $\approx 1.87$ times increase, and $1.74\%$ for retrained, an $\approx 4.46$ times increase).

## 5  FUTURE WORK

Our analysis of Langevin unlearning with asymmetric data sources provides deeper insights into the unlearning-utility trade-off and raises interesting research questions, particularly regarding appropriate unlearning assumptions for different problem settings. A natural extension involves studying Langevin unlearning in fine-tuning contexts, where public data is learned prior to incorporating private data. We also propose developing adaptive unlearning algorithms that optimally balance data alignment with unlearning efficiency by leveraging techniques from domain adaptation and differential privacy. Another promising direction is a constrained optimization approach to asymmetric machine unlearning that extends beyond retain set fine-tuning, where the objective minimizes loss on the retain set subject to the constraint that the unlearning weight distribution remains sufficiently close to a distribution trained exclusively on public data.

From a theoretical perspective, existing Langevin unlearning analysis in both mini-batch and full batch settings (Chien et al., 2024a) still suffers from intractable log-Sobolev constants. Alternative isoperimetric assumptions (Chewi et al., 2021; Mousavi-Hosseini et al., 2023; Altschuler & Chewi, 2024) or adopting weaker divergence measures could yield more tractable bounds. While Rényi divergence provides natural connections to differential privacy, machine unlearning presents distinct challenges that may benefit from relaxed theoretical assumptions. Finally, extending our analysis from weight distributions to output distributions would facilitate both evaluation and analysis, while staying relevant for black-box commercial models.

## 6  CONCLUSION

We have studied Langevin unlearning under the assumption of asymmetric data sources, where datasets contain both private and public data. Our theoretical analysis demonstrates that this framework fundamentally improves the unlearning-utility trade-off by enabling control over unlearning guarantees through data supplementation rather than noise amplification. The framework provides fine-grained analysis of how distributional alignment between public and private data affects this trade-off: when distributions are well-aligned, public data injection preserves utility while maintaining unlearning guarantees, while misaligned distributions introduce controlled performance penalties that remain more manageable than traditional noise-based approaches.

## 7 REPRODUCIBILITY STATEMENT

All theoretical results are supported by complete proofs in the Appendix (Theorems 3.1 to 3.3 in Appendices A.1, A.3 and A.4, respectively). Our anonymized codebase, including experimental scripts and configurations, is available at `https://anonymous.4open.science/r/asymmetric_langevin_unlearning-34A3` and `https://anonymous.4open.science/r/U-LiRAexperiments-EC08/`. All experiments settings are detailed in Appendix A.7 and Appendix A.8

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

# A  APPENDIX

## A.1  PROOF OF THEOREM 3.2

**Theorem.** *(Chien et al., 2024a) Suppose that the loss is $L$-smooth and $M$-Lipschitz, and that the learning distribution of weights at time $T$ satisfies a $C$ log-Sobolev inequality. Then, the Rényi divergence between $\pi_U^K$ (the unlearning distribution after $K$ iterations) and the retraining distribution after $T + K$ iterations is upper bounded by:*

$$D_\alpha(\pi_R^{T+K}\|\pi_U^K) \leq D_\alpha(\pi_L^T\|\pi_R^T)\exp\left(-\frac{1}{\alpha}\sum_{k=0}^{K-1}R_k\right)$$

*where $R_k > 0$ depend on the problem setting (Chien et al., 2024a). Moreover, if the loss function is $m$-strongly convex and the initial log-Sobolev constant satisfies $C > \frac{\sigma^2}{m}$, we get the following exponential decay of the Rényi divergence with respect to the unlearning iteration:*

$$D_\alpha(\pi_R^{T+K}\|\pi_U^K) \leq D_\alpha(\pi_L^T\|\pi_R^T)\exp\left(-\frac{2K\sigma^2\eta}{C\alpha}\right)$$

We provide the proof of (Chien et al., 2024a), Theorem 3.2, slightly modified to our setting. Specifically, we relax the assumption that the learning and retraining processes have converged to their stationary distribution (infinite training). In order to prove this theorem, we will use the following lemmas:

**Lemma A.1** (Characterizing the log-Sobolev constants of the PNGD updates (Chewi, 2023)). *Consider the PNGD update:*

$$\theta^{k+1} = \Pi_\Theta\left[\theta_k - \eta\nabla\mathcal{L}_D(\theta^k) + \sqrt{2\eta\sigma^2}W_k\right], \theta^0 \sim \pi$$

*where $\pi$ satisfies a $C$-Log Sobolev inequality. Then, we have the following:*

- *If $\mathcal{L}$ is $L$-smooth, then for the gradient update $h(\theta) = \theta - \nabla_\theta\mathcal{L}(\theta)$, we have that the distribution of $h_\sharp\pi$ satisfies a $(1 + \eta L)^2 \times C$ log-Sobolev inequality. Moreover, if $\mathcal{L}$ is $m$-strongly convex and $\eta < \frac{1}{L}$, then $h_\sharp\pi$ satisfies a $(1 - \eta m)^2 \times C$ log Sobolev inequality (Altschuler & Talwar, 2022).*

- *$\pi * \mathcal{N}(0, \sigma^2 I_d)$ satisfies a a $C + \sigma^2$ log-Sobolev inequality*

- *$\Pi_{\Theta\sharp}\pi$ satisfies a $C$ log-Sobolev inequality*

*By composing the aforementioned statements, we get that $\pi_1$ satisfies a $(1 + \eta L)^2 \times C + 2\eta\sigma^2$-log Sobolev inequality. Moreover, if $\mathcal{L}$ is $m$-strongly convex and $\eta < \frac{1}{L}$, we have that $\pi_1$ satisfies a $(1 - \eta m)^2 \times C + 2\eta\sigma^2$*

**Lemma A.2** (Data Processing inequality for the Rényi divergence (Erven & Harremoës, 2014)). *For any $\alpha \geq 1$, any function $h : \mathbb{R}^d \to \mathbb{R}^d$ and distributions $P, Q$ supported on $\mathbb{R}^d$, we have:*

$$D_\alpha(h_\sharp P\|h_\sharp Q) \leq D_\alpha(P\|Q)$$

*with equality if $h$ is bijective*

**Lemma A.3** ((Vempala & Wibisono, 2019; Chien et al., 2024a) characterizing the Rényi divergence between two distributions convoluted with Gaussians). *Let $P_t = P * \mathcal{N}(0, 2t\sigma^2 I_d)$ and $Q_t = Q * \mathcal{N}(0, 2t\sigma^2 I_d)$. Then, $\forall\alpha > 0$:*

$$\frac{\partial D_\alpha(P_t\|Q_t)}{\partial t} = -\alpha\sigma^2\frac{G_\alpha(P_t\|Q_t)}{F_\alpha(P_t\|Q_t)}$$

*with $G_\alpha(P\|Q) = \mathbb{E}_Q\left[\left(\frac{p}{q}\right)^\alpha\|\nabla\log\frac{p}{q}\|^2\right]$ denoting the relative Rényi information and $F_\alpha(P\|Q) = \mathbb{E}_Q\left[\left(\frac{p}{q}\right)^\alpha\right] = \exp((\alpha-1)D_\alpha(P\|Q))$*

**Lemma A.4.** *Lower bound of the G-F ratio (Vempala & Wibisono, 2019) If $Q \in \mathcal{P}(\Theta)$ satisfies a $C$ log Sobolev inequality, then $\forall P \in \mathcal{P}(\Theta)$:*

$$\frac{G_\alpha(P\|Q)}{F_\alpha(P\|Q)} \geq \frac{2D_\alpha(P\|Q)}{\alpha^2 C}$$

**Lemma A.5.** *Grönwall's inequality (Gronwall, 1919) Let $\mathbf{I} = [a, b]$ denote an interval on the real line. Let $\beta$ and $u$ be real-valued continuous functions defined on $\mathbf{I}$. If $u$ is differentiable in the interior of $\mathbf{I}$ and satisfies for all $t$ in the interior of $\mathbf{I}$:*

$$\frac{\partial u(t)}{dt} \leq \beta(t) u(t)$$

*then we have:*

$$u(t) \leq u(a) \exp\left(\int_a^t \beta(s) ds\right)$$

*for all $t \in I$*

**Lemma A.6.** *Universal upper bound on the log Sobolev constant for measures with compact support (Chen et al., 2021) Let $P$ a probability measure supported on a compact set with radius $R$. Then, for each $\sigma > 0$, $P * \mathcal{N}(0, \sigma I_d)$ satisfy a log Sobolev inequality with constant upper bounded by $6(4R^2 + \sigma) \exp\left(\frac{4R^2}{\sigma}\right)$*

*Proof.* Using these results, we have:

$$D_\alpha(h_\sharp \pi_R^{T+K} \| h_\sharp \pi_U^K) \leq D_\alpha(\pi_R^{T+K} \| \pi_U^K) \qquad (\text{ Lemma A.2})$$

**The PNGD updates preserve the log-Sobolev inequality for the resulting distributions:** let $\pi_U^{K,1,t} = h_\sharp \pi_U^K * \mathcal{N}(0, 2t\sigma^2 I_d)$ and $\pi_R^{T+K,1,t} = h_\sharp \pi_U^K * \mathcal{N}(0, 2t\sigma^2 I_d)$. Since $\pi_L^T$ and $\pi_R^T$ satisfy a log-Sobolev inequality (initialization distributions) and the loss function is $L$-smooth, then by Lemma A.1 the distributions $\pi_U^K, \pi_L^{T+K}$ satisfy respectively $C_{U,K}, C_{L,T+K}$ log Sobolev inequalities. Using Lemma A.1 on the distributions $\pi_U^{K,1,t}, \pi_R^{T+K,1,t}$ yields that they respectively satisfy $(1 + \eta L)^2 C_{U,K} + 2\eta\sigma^2$ and $(1 + \eta L)^2 C_{L,T+K} + 2\eta\sigma^2$ log Sobolev inequalities for all $t \in [0, \eta]$.

**Upper bounding the distributions convolved with Gaussian distributions:** Using Lemma A.3, we have that, $\forall \alpha > 0$:

$$\frac{\partial D_\alpha(\pi_R^{T+K,1,t} \| \pi_U^{K,1,t})}{\partial t} = -\alpha\sigma^2 \frac{G_\alpha(\pi_R^{T+K,1,t} \| \pi_U^{K,1,t})}{F_\alpha(\pi_R^{T+K,1,t} \| \pi_U^{K,1,t})}$$

and since $\pi_U^{K,1,t}$ satisfies a $C_{U,K,t} = (1 + \eta L)^2 C_{U,K} + 2t\sigma^2$ log-Sobolev inequality, we can use Lemma A.4 to upper bound the derivative of the Rényi divergence with respect to $t \in [0, \eta]$:

$$\frac{\partial D_\alpha(\pi_R^{T+K,1,t} \| \pi_U^{K,1,t})}{\partial t} \leq -\frac{2\sigma^2}{\alpha C_{U,K,t}} D_\alpha(\pi_R^{T+K,1,t} \| \pi_U^{K,1,t})$$

Thus, by Grönwall's inequality (Lemma A.5), we have $\forall t \in [0, \eta]$:

$$D_\alpha(\pi_R^{T+K,1,t} \| \pi_U^{K,1,t}) \leq D_\alpha(h_\sharp \pi_R^{T+K} \| h_\sharp \pi_U^K) \exp\left(\int_0^t -\frac{2\sigma^2}{\alpha C_{U,K,s}} ds\right)$$

$$\leq D_\alpha(h_\sharp \pi_R^{T+K} \| h_\sharp \pi_U^K) \exp\left(\int_0^t -\frac{2\sigma^2}{\alpha\left((1 + \eta L)^2 C_{U,K} + 2s\sigma^2\right)} ds\right)$$

$$\leq D_\alpha(\pi_R^{T+K} \| \pi_U^K) \exp\left(\int_0^t -\frac{2\sigma^2}{\alpha\left((1 + \eta L)^2 C_{U,K} + 2s\sigma^2\right)} ds\right)$$
$$(\text{Lemma A.2})$$

Computing the integral yields:

$$\int_0^t -\frac{2\sigma^2}{\alpha\left((1 + \eta L)^2 C_{U,K} + 2s\sigma^2\right)} ds = -\frac{1}{\alpha} \int_0^t \frac{2\sigma^2}{(1 + \eta L)^2 C_{U,K} + 2s\sigma^2} ds$$

$$= -\frac{1}{\alpha}\left[\log\left((1 + \eta L)^2 C_{U,K} + 2t\sigma^2\right) - \log\left((1 + \eta L)^2 C_{U,K}\right)\right]$$

$$= -\frac{1}{\alpha}\left[\log\left(1 + \frac{2t\sigma^2}{(1 + \eta L)^2 C_{U,K}}\right)\right]$$

Thus, by setting $t = \eta$, we get:

$$D_\alpha(\pi_R^{T+K,1,\eta} \| \pi_U^{K,1,\eta}) \leq \left(1 + \frac{2t\sigma^2}{(1+\eta L)^2 C_{U,K}}\right)^{\frac{-1}{\alpha}} D_\alpha(\pi_R^{T+K} \| \pi_U^K)$$

Finally, using the data processing inequality for the projection of PNGD and iterating over the number of unlearning iterations, we get:

$$D_\alpha(\pi_R^{T+K+1} \| \pi_U^{K+1}) \leq D_\alpha(\pi_R^{T+K,1,\eta} \| \pi_U^{K,1,\eta})$$

$$\leq \left(1 + \frac{2t\sigma^2}{(1+\eta L)^2 C_{U,K}}\right)^{\frac{-1}{\alpha}} D_\alpha(\pi_R^{T+K} \| \pi_U^K)$$

$$\leq D_\alpha(\pi_R^T \| \pi_L^T) \prod_{k=1}^K \left(1 + \frac{2t\sigma^2}{(1+\eta L)^2 C_{U,k}}\right)^{\frac{-1}{\alpha}}$$

$\square$

## A.2 TRACKING THE LOG-SOBOLEV CONSTANTS

For a generic, $L$-smooth non-convex loss function $\mathcal{L}$, one can derive the following recurrence relation, $\forall k \geq 1$ upper bounding the log-Sobolev constants:

$$C_1 \leq (1+\eta L)^2 C_0 + 2\eta\sigma^2 \qquad \text{(Lemma A.1)}$$
$$C_2 \leq (1+\eta L)^4 C_0 + (1+\eta L)^2 2\eta\sigma^2 + (1+\eta L)^2$$
$$\cdots$$
$$C_K \leq (1+\eta L)^{2K} C_0 + 2\eta\sigma^2 \sum_{k=0}^{K-1} (1+\eta L)^2$$
$$\leq (1+\eta L)^{2K} C_0 + 2\eta\sigma^2 \frac{(1+\eta L)^{2K} - 1}{(1+\eta L)^2 - 1} \qquad (5)$$

If we add the assumption that the loss is convex, then the map $h(\theta) = \theta - \eta\nabla_\theta \mathcal{L}(\theta)$ is 1-Lipschitz for $\eta < \frac{2}{L}$ (Hardt et al., 2016) and we can reduce $(1+\eta L)$ to 1 in the aforementioned bounds:

$$C_K \leq C_0 + 2K\eta\sigma^2 \qquad (6)$$

Finally, assuming $m$-strong convexity yields that the map $h(\theta)$ is $1 - \eta m$-Lipschitz, which allows for the following **contractive** recurrence on the log-Sobolev constants $\forall k \geq 1$ by setting $\eta < \frac{2}{m}(1 - \frac{\sigma^2}{mC_0})$ (Chien et al., 2024a):

$$C_k \leq (1-\eta m)^2 C_{k-1} + 2\eta\sigma^2 \leq C_{k-1}$$
$$C_k \leq (1-\eta m)^{2K} C_0 + 2\eta\sigma^2 \frac{(1-\eta m)^{2K} - 1}{(1-\eta m)^2 - 1} \leq C_0$$

Thus, we have that $\forall t \in [0, \eta]$, $\pi_U^{K,1,t}$ satisfies a $C_0$ log-Sobolev inequality thus we have by Lemma A.4:

$$\frac{\partial D_\alpha(\pi_R^{T+K,1,t} \| \pi_U^{K,1,t})}{\partial t} \leq -\frac{2\sigma^2}{\alpha C} D_\alpha(\pi_R^{T+K,1,t} \| \pi_U^{K,1,t})$$

Thus, by Grönwall's inequality (Lemma A.5), we have $\forall t \in [0, \eta]$:

$$\frac{\partial D_\alpha(\pi_R^{T+K,1,t} \| \pi_U^{K,1,t})}{\partial t} \leq D_\alpha(h_\sharp \pi_R^{T+K} \| h_\sharp \pi_U^K) \exp\left(\int_0^t -\frac{2\sigma^2}{\alpha C} ds\right)$$

$$\leq D_\alpha(h_\sharp \pi_R^{T+K} \| h_\sharp \pi_U^K) \exp\left(-\frac{2t\sigma^2}{\alpha C}\right)$$

Thus, by setting $t = \eta$ and using similar steps as the non convex proof above, we get the following result:

$$D_\alpha(\pi_R^{T+K}\|\pi_U^K) \leq D_\alpha(\pi_R^T\|\pi_L^T) \exp\left(-\frac{2K\eta\sigma^2}{\alpha C}\right)$$

The message conveyed by the strongly convex proof is that if we have a universal iteration independent upper bound on the log Sobolev constants at each timestep of the PNGD updates, then we could have a more meaningful upper bound on the Rényi divergence. The non convex Eq. (5) and convex Eq. (6) recurrence bounds are non contractive and iteration dependent, so they do not allow to establish a convergence rate for Theorem 3.2. This is where the projection step of PNGD comes in handy, as it allows to leverage the geometry of the set $\Theta$ to get a more informative bound:

**Lemma A.7** (Log Sobolev inequality on measures supported on a compact set (Chen et al., 2021), Corollary 1). *Let $\pi$ be a probability measure on $\mathbb{R}^d$ supported on a compact set $\Theta$ with radius $R \geq 0$. Then, for each $t \geq 0$, $\mu * \mathcal{N}(0, tI_d)$ satisfy a log sobolev inequality with constant $C$ controlled by:*

$$C \leq 6\left(4R^2 + t\right)\exp\left(\frac{4R^2}{t}\right)$$

**Proposition A.1** (Universal bound on the log Sobolev constants of distributions induced by PNGD updates (Chien et al., 2024a)). *Suppose that $\mathcal{L}$ is $M$ Lipschitz. Let $\theta_0 \sim \pi_0 \in \mathcal{P}(\Theta)$ where $\Theta$ is a compact set of radius $R$ and denote by $\pi_k$ the distribution $\theta_k$, the $k$-th iterate of PNGD (Eq. (2)). Then, $\forall k \geq 0$, $\pi_k$ satisfies a log-Sobolev inequality with constant $C_k$ controlled by:*

$$C_k \leq 6\left(4(R + \eta M)^2 + 2\eta\sigma^2\right)\exp\left(\frac{4(R + \eta M)^2}{2\eta\sigma^2}\right)$$

We can thus derive a similar bound to the strongly convex setting, for the non convex/convex settings:

Using Proposition A.1, we have $\forall k \geq 0$ that $\pi_U^K$ satisfies a $\tilde{C} = 6\left(4(R + \eta M)^2 + 2\eta\sigma^2\right)\exp\left(\frac{4(R+\eta M)^2}{2\eta\sigma^2}\right)$ log Sobolev inequality. Thus, using Lemma A.4, we have:

$$\frac{\partial D_\alpha(\pi_R^{T+K,1,t}\|\pi_U^{K,1,t})}{\partial t} \leq -\frac{2\sigma^2}{\alpha\tilde{C}}D_\alpha(\pi_R^{T+K,1,t}\|\pi_U^{K,1,t})$$

Thus, by Grönwall's inequality (Lemma A.5), we have $\forall t \in [0, \eta]$:

$$\frac{\partial D_\alpha(\pi_R^{T+K,1,t}\|\pi_U^{K,1,t})}{\partial t} \leq D_\alpha(h_\sharp\pi_R^{T+K}\|h_\sharp\pi_U^K)\exp\left(\int_0^t -\frac{2\sigma^2}{\alpha\tilde{C}}ds\right)$$

$$\leq D_\alpha(h_\sharp\pi_R^{T+K}\|h_\sharp\pi_U^K)\exp\left(-\frac{2t\sigma^2}{\alpha\tilde{C}}\right)$$

Finally, similarly to the strongly convex proofs, we can deduce that:

$$D_\alpha(\pi_R^{T+K}\|\pi_U^K) \leq D_\alpha(\pi_R^T\|\pi_L^T)\exp\left(-\frac{2K\eta\sigma^2}{\alpha\tilde{C}}\right)$$

## A.3 PROOF OF THEOREM 3.1

**Theorem.** *Suppose that the loss is $L$-smooth and $M$-Lipschitz, and that the initialization distribution satsifies a $C$-log Sobolev inequality. Moreover, suppose that the PNGD updates project onto a compact set $\Theta$ of radius $R$.*
*Then at learning iteration $T$, we have the following upper bound on the Renyi divergence between the retraining $\pi_R^T$ and learning $\pi_L^T$ distributions:*

$$\frac{D_\alpha(\pi_R^T\|\pi_L^T)}{\alpha} \leq \frac{2M^2\eta^2 n_{\text{forget}}^2}{(n_{\text{pub}} + n_{\text{priv}})^2\sigma^2}\sum_{t=1}^{T-1}\prod_{t'=t}^{T-1}\left(1 + \frac{\eta\sigma^2}{C_{t',1}}\right)^{-1}$$

*where $C_{t',1} > 0$ are log Sobolev constants of the distributions of the intermediate PNGD updates. Using the support's radius allows to loosely upper bound those constants (Chien et al., 2024a):*
*$C_{t',1} \leq 6e^{4\tau}(4\tau^2 + \eta\sigma^2)$ with $\tau = R + \eta M$*

*Proof.* The following proof is an adaptation of the proof of Theorem 3.2 in Chien et al. (2024a) to the asymmetric data setting.

Consider the following updates done during training. Recall that we are using full batch projected noisy gradient descent:

$$\theta_L^{t+1} = \Pi_\Theta \left[ \theta_L^t + \eta \nabla \mathcal{L}_{D_{\text{pub}} \cup D_{\text{priv}}}(\theta_L^t) + \sqrt{2\eta\sigma^2} W_t \right] \qquad (W_t \sim \mathcal{N}(0, I_d))$$

$$\theta_R^{t+1} = \Pi_\Theta \left[ \theta_R^t + \eta \nabla \mathcal{L}_{D_{\text{retain}}}(\theta_R^t) + \sqrt{2\eta\sigma^2} W_t \right] \qquad (W_t \sim \mathcal{N}(0, I_d))$$

Let's divide each optimization step into the following:

$$\theta_L^{t,1} = \theta_L^t + \eta \nabla \mathcal{L}_{D_{\text{pub}} \cup D_{\text{priv}}}(\theta_L^t) + \sqrt{\eta\sigma^2} W_t$$

$$\theta_R^{t,1} = \theta_R^t + \eta \nabla \mathcal{L}_{D_{\text{retain}}}(\theta_R^t) + \sqrt{\eta\sigma^2} W_t$$

Therefore, we can write

$$\theta_L^{t+1} = \Pi_\Theta \left[ \theta_L^{t,1} + \sqrt{\eta\sigma^2} W_t \right] \qquad (7)$$

$$\theta_R^{t+1} = \Pi_\Theta \left[ \theta_R^{t,1} + \sqrt{\eta\sigma^2} W_t \right]. \qquad (8)$$

Let $\pi_R^t, \pi_R^{t,1}, \pi_L^t, \pi_L^{t,1}$ be the distributions of respectively $\theta_R^t, \theta_R^{t,1}, \theta_L^t, \theta_L^{t,1}$

**The main question we try to tackle here is: what is $D_\alpha(\pi_R^t \| \pi_L^t)$ ?**

We first compare the distributions $\pi_R^{t,1}$ and $\pi_L^{t,1}$. By composition theorem of the Gaussian mechanism for Rényi Differential privacy (Mironov, 2017), and equivalently for the Rényi divergence, we have:

$$\frac{D_\alpha(\pi_R^{t,1} \| \pi_L^{t,1})}{\alpha} \leq \frac{D_\alpha(\pi_R^t \| \pi_L^t)}{\alpha} + \frac{\Delta_F^2}{2\sigma^2} \qquad (9)$$

where $\Delta_F$ is the $l_2$ sensitivity of the gradient update. For the next computations, let $n_{\text{pub}}$ denote the number of public points, $n_{\text{forget}}$ denote the number of points to forget, and $n_{\text{r-priv}}$ denote the number of *remaining* private points in the retain set. Computing the sensitivity in the asymmetric setting yields:

$$\Delta_F = \max_\theta \eta \| \nabla \mathcal{L}_{D_{\text{retain}}}(\theta) - \nabla \mathcal{L}_{D_{\text{pub}} \cup D_{\text{priv}}}(\theta) \|$$

$$= \max_\theta \eta \| \frac{1}{n_{\text{pub}} + n_{\text{r-priv}}} \sum_{d_i \in \text{I} \cup \text{II}} \nabla l(\theta, d_i) - \frac{1}{n_{\text{pub}} + n_{\text{r-priv}} + n_{\text{forget}}} \sum_{d_i \in \text{I} \cup \text{II} \cup \text{III}} \nabla \| l(\theta, d_i) \|$$

$$\leq \eta \left( \frac{1}{n_{\text{pub}} + n_{\text{r-priv}}} - \frac{1}{n_{\text{pub}} + n_{\text{r-priv}} + n_{\text{forget}}} \right) \sum_{d_i \in \text{I} \cup \text{II}} \| \nabla l(\theta, d_i) \|$$

$$+ \frac{\eta}{n_{\text{pub}} + n_{\text{r-priv}} + n_{\text{forget}}} \sum_{d_i \in \text{I} \cup \text{II} \cup \text{III}} \| \nabla l(\theta, d_i) \|$$

$$\leq M\eta(n_{\text{pub}} + n_{\text{r-priv}}) \left( \frac{1}{n_{\text{pub}} + n_{\text{r-priv}}} - \frac{1}{n_{\text{pub}} + n_{\text{r-priv}} + n_{\text{forget}}} \right) + \frac{n_{\text{forget}} M\eta}{n_{\text{pub}} + n_{\text{r-priv}} + n_{\text{forget}}}$$

$$\leq \underbrace{\frac{2M\eta n_{\text{forget}}}{n_{\text{pub}} + n_{\text{r-priv}} + n_{\text{forget}}}}_{\varepsilon}$$

**Lemma A.8.** *(Ye & Shokri, 2022) For any distributions $\xi_t, \xi_t'$ both satisfying $C_{t,1}$-LSI, we have:*

$$\frac{D_\alpha(\xi_t * \mathcal{N}(0, \eta\sigma^2 I), \xi_t' * \mathcal{N}(0, \eta\sigma^2 I))}{\alpha} \leq \frac{D_{\alpha(t)}(\xi_t, \xi_t')}{\alpha(t)} \left( 1 + \frac{\eta\sigma^2}{C_{t,1}} \right)^{-1}$$

*where $\alpha(t) = \frac{\alpha - 1}{1 + \frac{\eta\sigma^2}{C_{t,1}}}$*

By combining the data processing inequality (projection) and Lemma A.8, we get the following recurrence inequality:

$$\frac{D_\alpha(\pi_R^{T+1}\|\pi_L^{T+1})}{\alpha} \leq \left(\frac{D_{\alpha(T)}(\pi_R^T\|\pi_L^T)}{\alpha(T)} + \frac{\varepsilon^2}{2\sigma^2}\right)\left(1 + \frac{\eta\sigma^2}{C_{T,1}}\right)^{-1}$$

$$= \frac{\varepsilon^2}{2\sigma^2}\left(1 + \frac{\eta\sigma^2}{C_{T,1}}\right)^{-1} + \frac{D_{\alpha(T)}(\pi_R^T\|\pi_L^T)}{\alpha(T)}\left(1 + \frac{\eta\sigma^2}{C_{T,1}}\right)^{-1}$$

$$\leq \frac{\varepsilon^2}{2\sigma^2}\left(1 + \frac{\eta\sigma^2}{C_{T,1}}\right)^{-1} + \left(\frac{D_{\alpha(T-1)}(\pi_R^T\|\pi_L^T)}{\alpha(T-1)} + \frac{\varepsilon^2}{2\sigma^2}\right)\left(1 + \frac{\eta\sigma^2}{C_{T,1}}\right)^{-1}\left(1 + \frac{\eta\sigma^2}{C_{T-1,1}}\right)^{-1}$$

$$\leq \frac{\varepsilon^2}{2\sigma^2}\left[B(T) + B(T-1)\right] + B(T-2)\left(\frac{D_{\alpha(T-2)}(\pi_R^T\|\pi_L^T)}{\alpha(T-2)} + \frac{\varepsilon^2}{2\sigma^2}\right)$$

$$\text{(where } B(t) = \prod_{k=t}^T\left(1 + \frac{\eta\sigma^2}{C_{k,1}}\right)^{-1}\text{)}$$

$$\leq \frac{\varepsilon^2}{2\sigma^2}\sum_{i=1}^T B(i) + B(0)\left(\frac{D_{\alpha(0)}(\pi_R^T\|\pi_L^T)}{\alpha(0)} + \frac{\varepsilon^2}{2\sigma^2}\right)$$

$$\leq \frac{\varepsilon^2}{2\sigma^2}\sum_{i=0}^T B(i) \qquad\qquad \text{(since } D_{\alpha(t)}(\pi_0\|\pi_0) = 0\text{)}$$

$$= \frac{\varepsilon^2}{2\sigma^2}\sum_{t=0}^T\prod_{t'=t}^T\left(1 + \frac{\eta\sigma^2}{C_{t',1}}\right)^{-1}$$

The upper bound on the log Sobolev constants can be tracked in a similar fashion as in Proposition A.1 because of the projection onto the compact set $\Theta$. $\qquad\square$

### A.4   PROOF OF THEOREM 3.3

**Proposition.** *Assuming the data generating distributions share the same support, that the weight space $\Theta$ is compact and that the loss is $M$-Lipschitz wrt $\theta$, we have the following upper bound on the generalization error on the private data after performing $K$ iterations of unlearning, and initializing a weight $\theta_0$ from $\pi_L^T$:*

$$\mathbb{E}_{\theta\sim\pi_U}\left[\mathbb{E}_{x\sim P_{\text{priv}}}[\mathcal{L}(\theta,x)]\right] \leq \underbrace{\exp\left(\frac{n_{\text{pub}}}{n_{\text{pub}} + n_{\text{retain}}}D_\infty(P_{\text{priv}}\|P_{\text{pub}})\right)}_{\text{distribution mismatch penalty}}\mathbb{E}_{\theta\sim\pi_R}\left[\mathbb{E}_{d\sim P_{\text{train}}}[\mathcal{L}(\theta,d)]\right] +$$

$$M \times diam(\Theta) \times \underbrace{\sqrt{\frac{1}{2}D_\alpha(\pi_R\|\pi_U)}}_{\text{unlearning approximation error}}$$

*where $D_\infty(P\|Q) = \log\left(\text{ess sup}_{x\sim Q}\frac{p(x)}{q(x)}\right)$ is the infinite Rényi divergence (worst case regret (Erven & Harremoës, 2014)) and $p_{\text{train}}$ denotes the mixture of distributions $D_{\text{pub}}$ and $D_{\text{priv}}$ used for training the model.*

In order to prove Theorem 3.3, we will use the following quantities to define a set of preliminary lemmas.

#### A.4.1   PERFORMANCE ON THE TRAINING DISTRIBUTION MIXTURE

**Definition A.1** (Wasserstein distance). *The Wasserstein-1 distance is defined as*

$$W_1(\mu,\nu) = \inf_{\gamma\in\Pi(\mu,\nu)}\int_{\mathcal{X}\times\mathcal{X}} d(x,y)\, d\gamma(x,y),$$

*where:*

- $\mu$ and $\nu$ are probability measures on a metric space $(\mathcal{X}, d)$,

- $d(x, y)$ is the distance between points $x, y \in \mathcal{X}$,

- $\Pi(\mu, \nu)$ is the set of all couplings of $\mu$ and $\nu$, i.e., the set of joint distributions $\gamma$ on $\mathcal{X} \times \mathcal{X}$ such that the marginals of $\gamma$ are $\mu$ and $\nu$:

$$\int_{\mathcal{X}} \gamma(x, y)\, dy = \mu(x), \quad \int_{\mathcal{X}} \gamma(x, y)\, dx = \nu(y).$$

**Definition A.2** (Total Variation Distance). *Let $P$ and $Q$ be two probability measures on a measurable space $(\Omega, \mathcal{F})$. The **total variation distance** between $P$ and $Q$ is defined as*

$$TV(P, Q) = \sup_{A \in \mathcal{F}} |P(A) - Q(A)| \tag{10}$$

$$= \frac{1}{2} \int_{\Omega} |dP - dQ| \tag{11}$$

$$= \frac{1}{2} \|P - Q\|_{TV}. \tag{12}$$

**Theorem A.1.** *(Kantorovich Rubinstein's duality, (Villani et al., 2009), Theorem 5.10) If $\mu, \nu$ have a bounded support $\Omega$, then*

$$W_1(\mu, \nu) = \sup_{\|h\|_L \leq 1} \mathbb{E}_{x \sim \mu}[h(x)] - \mathbb{E}_{y \sim \nu}[h(y)], \tag{13}$$

*where $\|h\|_L \leq 1$ denotes the set of 1-Lipschitz functions on $\Omega$*

Let $f : \Theta \to \mathbb{R}$ such that $f(\theta) = \mathbb{E}_{D \sim P_{\text{train}}}[\mathcal{L}_D(\theta)]$, where $P_{\text{train}}$ denotes the training data distribution (a mixture of $P_{\text{priv}}$ and $P_{\text{pub}}$. Since $\mathcal{L}(., D)$ is $M-$Lipschitz, so is $f$. Then, we have that:

$$\mathbb{E}_{\substack{\theta \sim \pi_U \\ D \sim P_{\text{train}}}}[\mathcal{L}(\theta, D)] - \mathbb{E}_{\substack{\theta \sim \pi_R \\ D \sim P_{\text{train}}}}[\mathcal{L}(\theta, D)] = \mathbb{E}_{\theta \sim \pi_U}[f(\theta)] - \mathbb{E}_{\theta \sim \pi_R}[f(\theta)] \quad \text{(Fubini's theorem)}$$

$$\leq M \times W_1(\pi_U, \pi_R) \quad \text{(By Theorem A.1)}$$

Now, we need to find an upper bound on the 1-Wasserstein distance in terms of the Rényi divergence between $\pi_R$ and $\pi_U$. The following results will be useful in deriving it:

**Proposition A.2.** *(Pinsker's inequality) For two probability distributions $P, Q$, we have*

$$2TV(P, Q)^2 \leq KL(P||Q). \tag{14}$$

**Proposition A.3.** *(Monotonicity of Rényi divergence, (Erven & Harremoës, 2014)) For $1 \leq \alpha_1 \leq \alpha_2$ and probability measures $P, Q$,*

$$KL(P||Q) \leq D_{\alpha_1}(P||Q) \leq D_{\alpha_2}(P||Q).$$

*The KL lower bounds any Rényi divergence since it is obtained by the limit $\alpha \to 1$.*

**Proposition A.4.** *(Upper bounding $W_1$ with $TV$ (Gibbs & Su, 2002)) If the distributions $P, Q$ share a support $\Omega$ and $diam(\Omega) = \sup_{(x,y) \in \Omega \times \Omega} d(x, y)$ is finite, then we have*

$$W_1(P, Q) \leq diam(\Omega) TV(P, Q). \tag{15}$$

Using the results above, we have

$$\mathbb{E}_{\theta \sim \pi_U}[f(\theta)] - \mathbb{E}_{\theta \sim \pi_R}[f(\theta)] \leq M W_1(\pi_U, \pi_R)$$

$$\leq M \times diam(\Theta) \times TV(\pi_U, \pi_R)$$

$$\text{(By Proposition A.4 and compactness of } \Theta)$$

$$\leq M \times diam(\Theta) \times \sqrt{\frac{1}{2} KL(\pi_U, \pi_R)} \quad \text{(By Proposition A.3)}$$

$$\leq M \times diam(\Theta) \times \sqrt{\frac{1}{2} D_\alpha(\pi_U, \pi_R)} \quad \text{(By Proposition A.3)}$$

Thus, we obtain that the generalization error of learning + unlearning is upper bounded by:

**Proposition A.5.** *Assuming that $\mathcal{L}$ is $M$-Lipschitz, we have*

$$\mathbb{E}_{\theta \sim \pi_U}\left[\mathbb{E}_{D \sim P_{\text{train}}}[L(\theta, D)]\right] \leq \mathbb{E}_{\theta \sim \pi_R}\left[\mathbb{E}_{D \sim P_{\text{train}}}[L(\theta, D)]\right] + M \times diam(\Theta) \times \sqrt{\frac{1}{2} D_\alpha(\pi_U \| \pi_R)}$$

$$\tag{16}$$

### A.4.2 ADAPTING THE BOUND TO THEOREM 3.3

We would like to evaluate the performance of the model obtained after unlearning. Proposition A.5 provides a generalization bound on a mixture of distributions, namely on public data + private data. In most practical scenarios, one would want to quantify the "lost" performance on private data after forgetting one of its subsets. Thus, we would like to upper bound the quantity $\mathbb{E}_{\pi_U}\left[\mathbb{E}_{D\sim P_{\text{priv}}}[\mathcal{L}_D(\theta)]\right]$. The training data distribution used for either retraining or unlearning can be considered as generated from a mixture of the distributions $I$ and $II$. Assuming the sampling proportions for training are consistent, one can write that the data distribution used in retraining is

$$P_{\text{train}} = \frac{n_{\text{pub}}}{n_{\text{pub}} + n_{\text{r-priv}}} P_{\text{pub}} + \frac{n_{\text{r-priv}}}{n_{\text{pub}} + n_{\text{r-priv}}} P_{\text{priv}}.$$

Fix any $\theta \in \Theta$. We have that

$$\mathbb{E}_{\mathcal{D}\sim P_{\text{train}}}[\mathcal{L}(\theta,\mathcal{D})] = \frac{n_{\text{pub}}}{n_{\text{pub}} + n_{\text{r-priv}}} \mathbb{E}_{\mathcal{D}\sim P_{\text{pub}}}[\mathcal{L}(\theta,\mathcal{D})] + \frac{n_{\text{r-priv}}}{n_{\text{pub}} + n_{\text{r-priv}}} \mathbb{E}_{\mathcal{D}\sim P_{\text{priv}}}[\mathcal{L}(\theta,\mathcal{D})]$$

$$\begin{aligned}
\mathbb{E}_{\mathcal{D}\sim P_{\text{priv}}}[\mathcal{L}(\theta,\mathcal{D})] &= \int p_{\text{priv}}(x)\mathcal{L}(\theta,x)dx \\
&= \int p_{\text{train}}(x)\frac{p_{\text{priv}}(x)}{p_{\text{train}}(x)}\mathcal{L}(\theta,x)dx \\
&= \mathbb{E}_{x\sim P_{\text{train}}}\left[\frac{p_{\text{priv}}(x)}{p_{\text{train}}(x)}\mathcal{L}(\theta,x)\right] \\
&\leq \mathbb{E}_{d\sim P_{\text{train}}}[\text{ess sup}_{x\in Supp(P_{\text{pub}})\cup Supp(P_{\text{priv}})}\frac{p_{p_{\text{priv}}}(x)}{p_{\text{train}}(x)}\mathcal{L}(\theta,d)] \\
&\leq \text{ess sup}_{x\in Supp(P_{\text{pub}})\cup Supp(P_{\text{priv}})}\frac{p_{\text{priv}}(x)}{p_{\text{train}}(x)}\mathbb{E}_{d\sim P_{\text{train}}}[\mathcal{L}(\theta,d)] \\
&\leq \exp(D_\infty(P_{\text{priv}},P_{\text{train}}))\mathbb{E}_{d\sim P_{\text{train}}}[\mathcal{L}(\theta,d)].
\end{aligned}$$

Moreover, we have by convexity of the Rényi divergence (Erven & Harremoës, 2014) in its second argument that

$$D_\infty(P_{\text{priv}}\|P_{\text{train}}) \leq \frac{n_{\text{pub}}}{n_{\text{pub}} + n_{\text{r-priv}}}(P_{\text{priv}}\|P_{\text{pub}}).$$

Thus we also have

$$\mathbb{E}_{d\sim P_{\text{priv}}}[\mathcal{L}(\theta,d)] \leq \exp\left(\frac{n_{\text{pub}}}{n_{\text{pub}} + n_{\text{r-priv}}}D_\infty(P_{\text{priv}}\|P_{\text{pub}})\right)\mathbb{E}_{d\sim P_{\text{train}}}[\mathcal{L}(\theta,d)]. \quad (17)$$

Thus, we can adapt proposition A.5 to evaluate the risk *only* on private data. Note that so far, the only assumption made on the difference between the data generating distributions I and II is that they share the same support. The following bound might be refined with additional assumptions, such as covariate shift or conditional shift.

We can thus take the expectation of $\theta$ with respect to $\pi_U$ to get

$$\mathbb{E}_{\theta\sim\pi_U}\left[\mathbb{E}_{d\sim P_{\text{priv}}}[\mathcal{L}(\theta,d)]\right] \leq \exp\left(\frac{n_{\text{pub}}}{n_{\text{pub}} + n_{\text{r-priv}}}D_\infty(P_{\text{priv}}\|P_{\text{pub}})\right)\mathbb{E}_{\theta\sim\pi_U}\left[\mathbb{E}_{d\sim P_{\text{train}}}[\mathcal{L}(\theta,d)]\right],$$

and using proposition A.5 to upper bound $\mathbb{E}_{\theta\sim\pi_U}\left[\mathbb{E}_{d\sim P_{\text{train}}}[\mathcal{L}(\theta,d)]\right]$, we prove proposition 3.3:

$$\mathbb{E}_{\theta\sim\pi_U}\left[\mathbb{E}_{x\sim P_{\text{priv}}}[\mathcal{L}(\theta,x)]\right] \leq \exp\left(\frac{n_{\text{pub}}}{n_{\text{pub}} + n_{\text{retain}}}D_\infty(P_{\text{priv}}\|P_{\text{pub}})\right)\mathbb{E}_{\theta\sim\pi_R}\left[\mathbb{E}_{d\sim P_{\text{train}}}[\mathcal{L}(\theta,d)]\right] +$$

$$M \times diam(\Theta) \times \sqrt{\frac{1}{2}D_\alpha(\pi_R\|\pi_U)}.$$

---

**Algorithm 1** Training with Projected Noisy Gradient Descent (PNGD)

---

1: $\theta_0 \sim \pi_0$        ▷ Sample from initialization distribution
2: **for** $t = 0$ to $T - 1$ **do**
3:      $g_t \leftarrow \nabla_\theta L_D(\theta_t)$        ▷ Compute gradient on full dataset
4:      $\xi_t \sim \mathcal{N}(0, 2\eta\sigma^2 I_d)$        ▷ Sample Gaussian noise
5:      $\theta_{t+1} \leftarrow \Pi_\Theta[\theta_t - \eta g_t + \xi_t]$        ▷ Update and project
6: **end for**
7: **return** $\theta_T$

---

**Algorithm 2** Langevin Unlearning

---

1: $\theta_0^U \leftarrow \theta_T$        ▷ Initialize from trained model
2: **for** $k = 0$ to $K - 1$ **do**
3:      $g_k \leftarrow \nabla_\theta L_{D_{\text{retain}}}(\theta_k^U)$        ▷ Compute gradient on retain set only
4:      $\xi_k \sim \mathcal{N}(0, 2\eta\sigma^2 I_d)$        ▷ Sample Gaussian noise
5:      $\theta_{k+1}^U \leftarrow \Pi_\Theta[\theta_k^U - \eta g_k + \xi_k]$        ▷ Update and project
6: **end for**
7: **return** $\theta_K^U$

---

### A.5 LANGEVIN UNLEARNING PSEUDO-CODE

### A.6 DOMAINNET DATA

The following is a snippet of samples from the DomainNet dataset, where we extracted two domains, Clipart and Quickdraw. The classes are aggregated into 24 meta-classes Table 3, following (Peng et al., 2019).

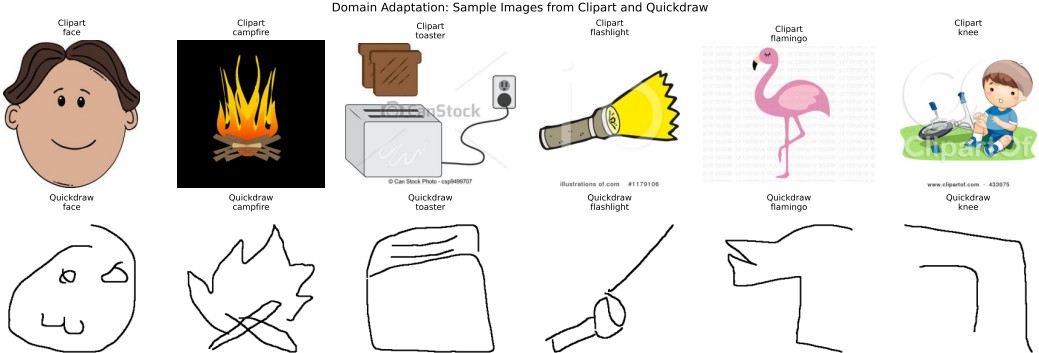

Figure 4: The two domains of public and private data used for Sections 4.1 and 4.2 (Peng et al., 2019). Both datasets share the same number of classes, with Clipart being a collection of stylized images representing the private data, and Quickdraw representing a collection of hand-draw sketches.

### A.7 DETAILS ABOUT THE RÉNYI ESTIMATION

#### A.7.1 NEURAL RÉNYI ESTIMATION

Following the works of Birrell et al. (2021; 2023), two variational representations of the Rényi divergence between two distributions $P, Q$ have been proposed. The first draws inspiration from the Donsker–Varadhan dual representation (Donsker & Varadhan, 1975) of the KL divergence:

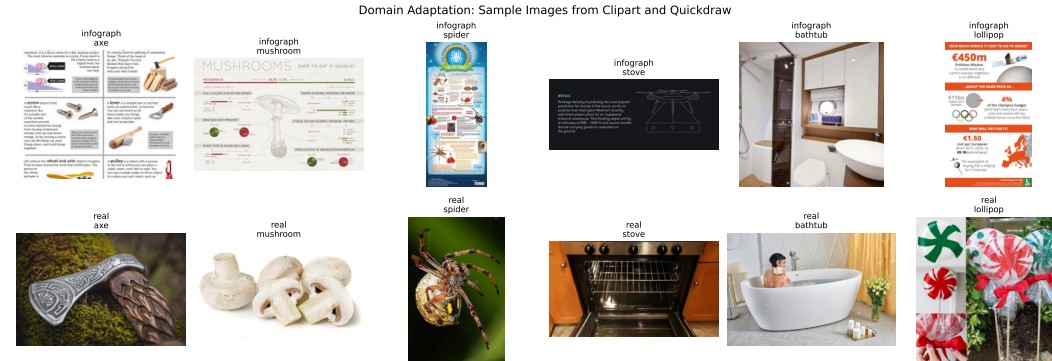

Figure 5: The two domains of public and private data used for Section 4.2 (Peng et al., 2019). Both datasets share the same number of classes, with Infograph being a collection of stylized images representing the public data, and Real representing a collection of real-life images.

**Theorem A.2** (Donsker–Varadhan Rényi divergence (Birrell et al., 2021)). *Let $P, Q$ be two distributions on $(\Omega, \mathcal{M})$ and $\alpha \in \mathbb{R}$, $\alpha \neq 0, 1$. Then, for any set of functions $\Phi$ with $\mathcal{M}_b(\Omega) \subset \Phi \subset \mathcal{M}(\Omega)$,*

$$\frac{D_\alpha(P\|Q)}{\alpha} = \sup_{\phi \in \Phi} \left\{ \frac{1}{\alpha - 1} \log \int e^{(\alpha-1)\phi} \, dP - \frac{1}{\alpha} \log \int e^{\alpha\phi} \, dQ \right\}. \quad (18)$$

*If in addition $(\Omega, \mathcal{M})$ is a metric space with the Borel $\sigma$-algebra, then Eq. (18) holds for all $\Phi$ satisfying $\mathrm{Lip}_b \subset \Phi \subset \mathcal{M}(\Omega)$, where $\mathrm{Lip}_b$ denotes the set of bounded Lipschitz functions.*

Here, $\mathcal{M}(\Omega)$ denotes the space of measurable real-valued functions on $\Omega$, and $\mathcal{M}_b(\Omega)$ the subspace of bounded functions.

While this representation allows sample-based estimation, it involves exponential terms that yield high-variance estimates in practice. To mitigate this issue, Birrell et al. (2023) proposed a convex conjugate formulation:

**Theorem A.3** (Convex conjugate Rényi divergence (Birrell et al., 2023)). *Let $P, Q$ be probability distributions supported on $\Omega$, with $P \ll Q$, and let $\mathcal{M}_b(\Omega)$ denote the space of bounded measurable functions. Then, for all $\alpha \in (0, +\infty) \setminus \{1\}$,*

$$\frac{D_\alpha(P\|Q)}{\alpha} = \sup_{g \in \mathcal{M}_b(\Omega),\, g<0} \int g \, dQ + \frac{1}{\alpha - 1} \int |g|^{\frac{\alpha-1}{\alpha}} \, dP + \frac{1}{\alpha}(\log \alpha + 1). \quad (19)$$

This convex conjugate formulation removes the exponential dependence and provides more stable numerical estimates, making it preferable for our setting.

**Neural network parameterization.** To approximate $\Phi = \{g \in \mathcal{M}(\Theta) : g < 0\}$ we use the class $g_\theta$ of two-layer MLPs with spectral normalization (Miyato et al., 2018), LeakyReLU activations, and a polysoftplus output activation as in Birrell et al. (2023). The polysoftplus activation offers superior numerical stability compared to ReLU. It is defined as

$$\mathrm{polysoftplus}(x) = -\left( \frac{1}{1-x} \mathbf{1}_{x<0} + (1+x)\mathbf{1}_{x\geq 0} \right). \quad (20)$$

The discriminator network $g_\theta$ is trained to maximize the variational bound in Eq. (4) using samples $\{\theta_i^U\}_{i=1}^N \sim \pi_U^K$ and $\{\theta_j^R\}_{j=1}^N \sim \pi_R^{T+K}$. The optimization objective becomes:

$$\max_\theta \left\{ \frac{1}{N} \sum_{j=1}^N g_\theta(\theta_j^R) + \frac{1}{\alpha-1} \frac{1}{N} \sum_{i=1}^N |g_\theta(\theta_i^U)|^{\frac{\alpha-1}{\alpha}} + \frac{1}{\alpha}(\log \alpha + 1) \right\}. \quad (21)$$

To reduce estimator variance, we repeat the discriminator training five times with different random initializations and report the average. We use a learning rate of value $0.0001$ with Adam optimizer (Kingma & Ba, 2017), and train the discriminators for 30000 epochs with batch size $b = 6000$.

Table 3: Class aggregation for experimental dataset. Individual classes are grouped into 24 superclasses.

| Superclass | Individual Classes |
|---|---|
| Furniture | bathtub, bed, bench, ceiling fan, chair, chandelier, couch, door, dresser, fence, fireplace, floor lamp, hot tub, ladder, lantern, mailbox, picture frame, pillow, postcard, see saw, sink, sleeping bag, stairs, stove, streetlight, suitcase, swing set, table, teapot, toilet, toothbrush, toothpaste, umbrella, vase, wine glass |
| Mammal | bat, bear, camel, cat, cow, dog, dolphin, elephant, giraffe, hedgehog, horse, kangaroo, lion, monkey, mouse, panda, pig, rabbit, raccoon, rhinoceros, sheep, squirrel, tiger, whale, zebra |
| Tool | anvil, axe, bandage, basket, boomerang, bottlecap, broom, bucket, compass, drill, dumbbell, hammer, key, nail, paint can, passport, pliers, rake, rifle, saw, screwdriver, shovel, skateboard, stethoscope, stitches, sword, syringe, wheel |
| Cloth | belt, bowtie, bracelet, camouflage, crown, diamond, eyeglasses, flip flops, hat, helmet, jacket, lipstick, necklace, pants, purse, rollerskates, shoe, shorts, sock, sweater, t-shirt, underwear, wristwatch |
| Electricity | calculator, camera, cell phone, computer, cooler, dishwasher, fan, flashlight, headphones, keyboard, laptop, light bulb, megaphone, microphone, microwave, oven, power outlet, radio, remote control, spreadsheet, stereo, telephone, television, toaster, washing machine |
| Building | The Eiffel Tower, The Great Wall, barn, bridge, castle, church, diving board, garden, garden hose, golf club, hospital, house, jail, lighthouse, pond, pool, skyscraper, square, tent, waterslide, windmill |
| Office | alarm clock, backpack, binoculars, book, calendar, candle, clock, coffee cup, crayon, cup, envelope, eraser, map, marker, mug, paintbrush, paper clip, pencil, scissors |
| Human Body | arm, beard, brain, ear, elbow, eye, face, finger, foot, goatee, hand, knee, leg, moustache, mouth, nose, skull, smiley face, toe, tooth |
| Road Transportation | ambulance, bicycle, bulldozer, bus, car, firetruck, motorbike, pickup truck, police car, roller coaster, school bus, tractor, train, truck, van |
| Food | birthday cake, bread, cake, cookie, donut, hamburger, hot dog, ice cream, lollipop, peanut, pizza, popsicle, sandwich, steak |
| Nature | beach, cloud, hurricane, lightning, moon, mountain, ocean, rain, rainbow, river, snowflake, star, sun, tornado |
| Cold Blooded | crab, crocodile, fish, frog, lobster, octopus, scorpion, sea turtle, shark, snail, snake, spider |
| Music | cello, clarinet, drums, guitar, harp, piano, saxophone, trombone, trumpet, violin |
| Fruit | apple, banana, blackberry, blueberry, grapes, pear, pineapple, strawberry, watermelon |
| Sport | baseball, baseball bat, basketball, flying saucer, hockey puck, hockey stick, snorkel, soccer ball, tennis racquet, yoga |
| Tree | bush, cactus, flower, grass, house plant, leaf, palm tree, tree |
| Bird | bird, duck, flamingo, owl, parrot, penguin, swan |
| Vegetable | asparagus, broccoli, carrot, mushroom, onion, peas, potato, string bean |
| Shape | circle, hexagon, line, octagon, squiggle, triangle, zigzag |
| Kitchen | fork, frying pan, hourglass, knife, lighter, matches, spoon, wine bottle |
| Water Transportation | aircraft carrier, canoe, cruise ship, sailboat, speedboat, submarine |
| Sky Transportation | airplane, helicopter, hot air balloon, parachute |
| Insect | ant, bee, butterfly, mosquito |
| Others | The Mona Lisa, angel, animal migration, campfire, cannon, dragon, feather, fire hydrant, mermaid, snowman, stop sign, teddy-bear, traffic light |

This procedure used $N = 30,000$ model samples, which makes it computationally intensive and better suited for theoretical validation than for large-scale empirical benchmarking. Although regularization and repeated runs alleviate variance, Rényi divergence estimation remains a statistically challenging task. Developing scalable and lower-variance estimators is therefore an important direction for future work.

### A.7.2 SAMPLING FROM $\pi_U^K$ AND $\pi_R^{T+K}$

We conduct experiments on the DomainNet dataset (24-class image classification) Fig. 4. We choose the domain Clipart as the private data domain, which are stylized images, and Quick-draw, a collection of hand-drawn sketches as the public domain. Image embeddings are extracted using DinoV2 (Oquab et al., 2024), a self-supervised vision transformer. We specifically use vit_small_patch16_224_dino (Caron et al., 2021). All images are resized to $224 \times 224$ prior to feature extraction.

On these embeddings, we train 30,000 linear classifiers on the full dataset $D = D_{\text{pub}} \cup D_{\text{priv}}$ for $T = 20$ iterations, and subsequently fine-tune them on the retain set $D_r = D \setminus D_{\text{forget}}$ for $K \in \{1, 5, 10, 15\}$ additional iterations. This procedure yields 30,000 samples from the unlearning distribution $\pi_U^K$.

For comparison, we train another 30,000 linear classifiers directly on the retain set $D_r$ for $T + K$ iterations, producing samples from the retraining distribution $\pi_R^{T+K}$. All models are trained using the same projected noisy gradient descent (PNGD) update with noise scale $\sigma = 0.01$, learning rate $\eta = 0.001$, batch size $b = 1024$, and radius $R = 1.0$ using SGD.

To assess robustness across dataset splits, we fix the total training set size to $N_{\text{train}} = 42,000$, and vary the public and forget set sizes as $(|D_{\text{pub}}|, |D_{\text{forget}}|) \in \{(10,000, 12,000), (15,000, 7,000),$ and $(20,000, 2,000)\}$. The remaining private data in the retain set is fixed to have size 20,000. The resulting divergence estimates are reported in Figs. 2a and 2b.

### A.7.3 PSEUDO-CODE

---

**Algorithm 3** Rényi Divergence Estimation via Variational Representation

---

1: **Input:** Samples $\{\theta_i^R\}_{i=1}^N \sim \pi_R^{T+K}$, $\{\theta_j^U\}_{j=1}^N \sim \pi_U^K$, order $\alpha$, discriminator architecture
2: Initialize discriminator network $g_\phi$ with spectral normalization
3: **for** epoch = 1 to num_epochs **do**
4:    Sample minibatch from retraining samples $\{\theta_i^R\}$
5:    Sample minibatch from unlearning samples $\{\theta_j^U\}$
6:    Compute variational objective:

$$\mathcal{L} = \frac{1}{N}\sum_{i=1}^N g_\phi(\theta_i^R) + \frac{1}{\alpha - 1}\frac{1}{N}\sum_{j=1}^N |g_\phi(\theta_j^U)|^{\frac{\alpha-1}{\alpha}} + \frac{1}{\alpha}(\log \alpha + 1) \tag{22}$$

7:    Update $\phi$ to maximize $\mathcal{L}$ via gradient ascent
8: **end for**
9: **Output:** Estimated divergence $\widehat{D}_\alpha(\pi_U^K \| \pi_R^{T+K}) = \widehat{\mathcal{L}}^{1/\alpha}$

---

### A.8 EVALUATION WITH U-LiRA

#### A.8.1 U-LiRA DETAILS

U-LiRA, introduced by Hayes et al. (2025) as an adaptation of the LiRA membership inference attack (Carlini et al., 2021) to the unlearning setting, formalizes unlearning evaluation as a binary hypothesis test. The goal is to distinguish between two distributions over model parameters: the unlearning distribution $\pi_U^K$, obtained by training on the full dataset and subsequently applying the target unlearning algorithm to remove the influence of the forget set, and the retraining distribution $\pi_R^{T+K}$, obtained by training from scratch without the forget set. Letting $P(\theta \mid \cdot)$ denote the likelihood of observing model parameters $\theta$ under a given distribution, the Neyman–Pearson lemma (Neyman & Pearson, 1933) implies that the most powerful test for this discrimination problem is achieved by thresholding the likelihood ratio

$$\frac{P(\theta|\pi_U^K)}{P(\theta|\pi_R^{T+K})}$$

for model parameters $\theta$.

Since directly computing $P(\theta \mid \pi_U^K)$ and $P(\theta \mid \pi_R^{T+K})$ is infeasible in practice, U-LiRA employs a series of approximations. First, the two distributions are approximated empirically by sampling: the adversary trains $N$ models under $\pi_U^K$ (full training followed by unlearning) and $N$ models under $\pi_R^{T+K}$ (training from scratch without the forget set).

To reduce the sample complexity required for a low-variance estimate, U-LiRA projects models into a one-dimensional representation space via a statistic $f : \Theta \to \mathbb{R}$ (since we only run the attack on forget sets of size 1, we follow Hayes et al. (2025) and choose $f$ to be the model's confidence score

on the forget example, rescaled by the logit function $\phi(\omega) = \ln\left(\frac{\omega}{1-\omega}\right)$). The test is then conducted on the surrogate likelihood ratio

$$\frac{P(f(\theta)|f(\pi_U^K))}{P(f(\theta)|f(\pi_R^{T+K}))}.$$

As a final simplifying approximation, U-LiRA models the projected distributions as Gaussians

$$f(\pi_U^K) \approx \mathcal{N}(\mu_U, \sigma_U^2), \quad f(\pi_R^{T+K}) \approx \mathcal{N}(\mu_R \sigma_R^2),$$

where the parameters $(\mu_U, \sigma_U^2)$ and $(\mu_R, \sigma_R^2)$ are estimated directly from the $N$ sample models of each distribution.

In 4.3, we presented the attack through the lens of Bayes' rule (following Algorithm 1 of Hayes et al. (2025)), providing a more intuitive explanation for readers less familiar with hypothesis testing concepts.

### A.8.2 EXPERIMENTAL SETUP

We evaluate unlearning in binary sentiment classification of IMDB reviews (Maas et al., 2011), with Amazon product reviews (Zhang et al., 2015) as public data. Models are 2-layer LSTMs (Hochreiter & Schmidhuber, 1997), trained to minimize cross-entropy loss with projected noisy gradient descent (Gaussian noise variance $\sigma^2 = 0.01$, projection onto an $\ell_2$ ball of radius 100).

For each trial, the forget set consists of a 100 datapoints sampled uniformly from the IMDB reviews dataset. Following the U-LiRA framework, we generate 75 model samples from two distributions:

- **Unlearning distribution** $\pi_U^K$: models trained on 25,000 private datapoints plus the forget set for $T$ epochs, then finetuned without the forget set for $K$ epochs.
- **Retraining distribution** $\pi_R^{T+K}$: models trained from scratch on the same 25,000 private datapoints (excluding the forget set) for $T + K$ epochs.

We repeat this sampling process both with and without the inclusion of the 50,000 public datapoints during training and unlearning.

