# OpenReview forum: "Unlearning with asymmetric sources: improved unlearning-utility trade-off with public data"
_ICLR.cc/2026/Conference — Submitted to ICLR 2026_

### Official Review · Reviewer_gRLK · 2025-10-26

**Soundness:** 3
**Presentation:** 4
**Contribution:** 3
**Rating:** 8
**Confidence:** 3

**Summary:**

This paper investigates machine unlearning in a practical "asymmetric" setting, where the training dataset comprises both private data (subject to unlearning requests) and public data (always retained). The authors focus on Langevin Unlearning, a noisy gradient-based method, and demonstrate both theoretically and empirically that leveraging public data can significantly improve the fundamental trade-off between unlearning efficacy and model utility.

The core theoretical contribution is showing that the volume of public data reduces the Renyi divergence between the unlearned model distribution and the ideal retrained distribution (generated by retraining from scratch). This provides a new mechanism for achieving certified unlearning, relying on "data composition" rather than "noise amplification," which typically degrades utility.

Empirical validation is provided through two distinct methods: a direct estimation of Renyi divergence on a vision task, which confirms the theoretical scaling, and a standard membership inference attack, which confirms the practical unlearning efficacy.

**Strengths:**

1. The paper tackles a highly practical and important problem. As models are increasingly trained on mixed-sensitivity data, methods that can leverage this asymmetry are essential.

2. The paper provides a clear story for its core claim: public data makes the initial model and the retrained model closer in distribution (Thm 3.1), and this favorable starting point makes the final unlearning process more efficient and effective (Thm 3.2).

3. Theorem 3.3 provides a formal, interpretable bound that explicitly links post-unlearning utility to the distributional mismatch ($D_{\infty}(P_{priv} || P_{pub})$), which is an important factor in practice.

4. The claims are supported by two different experimental approaches (direct divergence estimation and MIA) on two different data modalities, which strengthens the paper's conclusions.

**Weaknesses:**

1. A finding in Table 1 is that model utility was preserved almost perfectly, despite a significant misalignment between the public and private  domains. The authors correctly note this implies their utility bound (Thm 3.3) is "overly conservative." The paper would be stronger if it discussed this interesting gap more deeply. Is this an artifact of the pre-trained embeddings (DinoV2) used, which might already align these domains?

2.  As mentioned, the theoretical novelty lies in the application of existing analysis tools to this new, asymmetric setting, rather than in the development of new theoretical machinery. This is a mild weakness, as the application itself is insightful, but it means the paper is more of a strong application paper than a fundamental theory paper.

**Questions:**

1. Regarding the "overly conservative" utility bound and the strong empirical utility results in Table 1: Do the authors have a hypothesis for why the utility was so high even with the Clipart/Quickdraw misalignment? Could this be related to the use of pre-trained DinoV2 embeddings?

2. The theoretical analysis is based on full-batch Projected Noisy Gradient Descent. How do the authors anticipate these results, particularly the quadratic scaling in Theorem 3.1, would transfer to the more practical mini-batch setting?

3. The experiments for Rényi estimation use $\alpha=2$. Why not numerically optimize over it and pick the lowest one?

---

> ### Author Response · Authors · 2025-11-21
>
> We thank the reviewer for the interesting observations. Please find below our thoughts and answers to the questions:
>
> 1- The gap between the theoretical bound's pessimism and Table 1's empirical utility is indeed noteworthy.
> To validate whether the strong empirical performance generalizes beyond pre-trained embeddings, we have added experiments in Section 4.3 using LSTMs trained from scratch on IMDB sentiment classification. We still see some consistency in the accuracies in this case, which is an interesting phenomena!
>
> 2- The reviewer raises a valid concern about the gap between our full-batch analysis and practical mini-batch training. Our theoretical results are derived for full-batch projected noisy gradient descent, and extending them to the mini-batch regime is an important direction. We note that Chien et al. (2024b) in "Stochastic Gradient Langevin Unlearning" provides a thorough treatment of mini-batch PNGD with convergence analysis. We would expect a slowdown in convergence, as demonstrated in their work.
>
> 3- Renyi divergence is monotonous wrt its order $\alpha$. Therefore, an optimal value would be $\alpha=0$ (reverse KL), but that provides *weaker* privacy guarantees. We wanted to provide a numerical estimation method that would be robust to higher orders.

---

### Official Review · Reviewer_cRhB · 2025-10-29

**Soundness:** 2
**Presentation:** 1
**Contribution:** 2
**Rating:** 2
**Confidence:** 3

**Summary:**

This paper investigates machine unlearning in a setting with both public and private data sources. The authors use Langevin Unlearning and theoretically argue that adding public data can improve the unlearning-utility trade-off. They provide a bound showing that public data volume quadratically reduces the Rényi divergence between the unlearning and retraining distributions, suggesting unlearning can be achieved with less noise, thereby preserving utility.

**Strengths:**

1. The central idea of using asymmetric data sources (public and private) to improve the unlearning-utility trade-off is timely and relevant.

2. The paper attempts to provide a theoretical grounding for this idea by analyzing Langevin Unlearning and deriving bounds based on Rényi divergence, which is a rigorous approach.

**Weaknesses:**

1. **Disconnect from Premise:** A core theoretical issue is the symmetric role that n_pub (public data size) and n_priv (private data size) seem to play in the main theorem (Theorem 3.1). The analysis suggests the same benefit could be achieved by simply adding more private data. This appears to contradict the paper's central premise, which is that leveraging *asymmetric* sources provides a unique benefit that is different from just having a larger private dataset.

2. **Potentially Vacuous Utility Bound:** The main utility trade-off bound appears potentially vacuous, particularly due to the term that increases exponentially with the max divergence (!) between the public and private distributions. This is a major concern. The only scenario where this bound seems non-vacuous is if this divergence is extremely small or zero. However, if the distributions are nearly identical, the theoretical contribution feels less significant, as the public data is effectively just more private data (tying back to the first point).

3. **Missing Comparisons & Restrictive Assumptions:** The paper's contribution is difficult to situate without a comparison to key recent work, such as Koloskova et al. (2025) "Certified Unlearning for Neural Networks", which reportedly offers certified unlearning with fewer assumptions. The assumptions made in this paper (e.g., log-sobolev) are quite restrictive and may severely limit the practical applicability of the theoretical results.

4. **Gap in Experimental Validation:** There is a disconnect between the theoretical claims for unlearning and the main experimental setup. The experiment uses a linear model on static embeddings, not a modern, non-convex deep neural network. It's highly unclear if any of the observed trade-offs (e.g., in Table 1) would hold in a more complex and realistic setting. The reported requirement of training 30,000 models to get this result also raises questions about the method's practicality.

5. **Weak Link Between Theory and Practice:** Following the previous point, the link between the theoretical values and the experimental parameters is weak. For instance, the noise level chosen is low, but the corresponding privacy guarantee (epsilon) is never calculated or reported. This makes it impossible to judge if the "unlearning" is meaningful (epsilon could be extremely large). Furthermore, for the utility experiments in Section 4.1, Membership Inference Attack (MIA) results are surprisingly absent from the main body for this first experiment, which is a standard way to empirically validate unlearning.

6. **Limited Empirical Results:** The empirical evidence presented is not fully convincing. In Figure 2a, for example, the confidence intervals are large and appear to overlap significantly. This makes it difficult to draw firm conclusions from the experiment and weakens the empirical support for the paper's claims.

**Questions:**

1. Could you elaborate on the symmetric roles of n_pub and n_priv in your theoretical results? If they are symmetric, how does this support the core claim of an asymmetric benefit, especially given the utility bound's exponential dependence on the max divergence?

2. What is the effective epsilon privacy guarantee for the noise level used in your experiments? And why were Membership Inference Attack (MIA) results for the Section 4.1 experiments not included in the main paper?

3. Can you justify the choice of a linear model on static embeddings for the main experiment? How do you expect these results to translate to non-convex deep learning models, which are the standard for most modern unlearning discussions?

4. How does your Langevin Unlearning approach compare to Koloskova et al. (2025) "Certified Unlearning for Neural Networks"? A discussion on this and the necessity of the strong log-sobolev assumption would be very helpful for positioning your contribution and the corresponding assumptions.

---

> ### Author Response · Authors · 2025-11-21
>
> We thank the reviewer for the insightful comments. Please find below our answers and corresponding modifications:
>
> **Symmetric roles :** (Q1 and W1)
>
> Theorem 3.1 demonstrates that total dataset size improves initialization, which is intuitive. The novelty of our approach can be explained as follows: (1) pragmatically, public data is abundant and acquisition-free, whereas expanding private datasets faces regulatory and practical constraints; (2) theoretically, Theorem 3.3 shows that even misaligned public data remains beneficial—a distinction that would not emerge from simply adding more private data. Prior Langevin-based unlearning could  control the noise magnitude $\sigma$, which directly degrades utility as it increases. Our framework introduces $n_{pub}$ as an additional knob for the unlearning-utility trade-off. Theorem 3.3 enables a characterization of when increasing public data is jointly beneficial for both unlearning efficacy and utility preservation.
>
>
> **Q4 and W3**:
> We clarify how our work relates to Koloskova et al. (2025) and address the log-Sobolev assumption.
> Our analysis improves upon Chien et al. (2024) by requiring the log-Sobolev inequality (LSI) only at initialization. The initialization distribution—typically Gaussian—naturally satisfies LSI, and by Lemma A.1, this property is preserved through PNGD iterations due to the smoothness of the loss function. Unlike prior work, we do not assume the stationary distribution satisfies LSI.
>
> Koloskova et al. (2025) requires indeed fewer assumptions (no smoothness assumptions on the loss function), making their approach applicable to deep networks. However, their bounds are data-agnostic, depending on projection set geometry and clipping constants rather than the structure of the training data. Our framework is different: we derive data-dependent bounds to leverage the abundance of public data in improving unlearning without noise amplification. This data-dependent perspective is essential to our contribution and incompatible with Koloskova et al.'s data-agnostic approach. We have added discussion of this distinction in the Related Works section and apologize for the initial omission.
>
> **Renyi estimation (W4 and Q3):
> Section 4.3 validates unlearning on LSTM-based sentiment classification —a non-convex, realistic setting. We have clarified this in the revised manuscript, directly addressing concerns about modern architectures.
> Section 4.1 serves a different purpose: enabling empirical investigation of the Rényi divergence between parameter distributions. This variational estimation approach is novel to both unlearning and differential privacy literature. The high sample complexity (30,000 models) reflects that this is a methodological contribution, not a practical audit tool. Its goal is to validate how the Rényi divergence behaves with increasing public data and unlearning iterations—testing Theorem 3.2's predictions. We cannot study the theoretical bounds empirically because the log-Sobolev constants in Theorem 3.2 are intractable to estimate. The variational approach provides a tractable way to study actual divergence behavior, circumventing these challenges.
>
>
> **epsilon budget** (W5 and Q2)
> Computing explicit epsilon-delta bounds for Langevin Unlearning in non-trivial settings is a known challenge in the literature. As demonstrated in prior work by Chien et al. (2024), deriving concrete privacy budgets requires knowledge of the loss function's smoothness constant and log-Sobolev constants, which are intractable to estimate in non-convex settings. Existing results that report epsilon are limited to toy problems like logistic regression. We instead directly validate our theoretical predictions through empirical divergence estimation in Section 4.1, where we numerically compute the Rényi divergence and demonstrate that it decreases monotonically with both public data volume and unlearning iterations.
> Regarding membership inference attacks, Section 4.1 focuses on parameter-space divergence estimation as a novel evaluation methodology. However, we recognize the importance of standard unlearning evaluation practices. We have added comprehensive membership inference attack experiments to Section 4.3, which provides MIA-based validation alongside our divergence estimates.

---

### Official Review · Reviewer_DwwT · 2025-10-31

**Soundness:** 3
**Presentation:** 3
**Contribution:** 2
**Rating:** 2
**Confidence:** 4

**Summary:**

This proof-based paper investigates the theoretical foundations of machine unlearning by analyzing the role of public and private data in achieving effective unlearning. The authors derive formal guarantees showing that as the amount of public data increases, the Rényi divergence between the retrained and unlearned model distributions decreases quadratically. This result highlights the critical utility of public data in approximating the retraining outcome without full retraining. Furthermore, the paper provides a detailed theoretical analysis of how distributional mismatch between public and private datasets influences the efficiency and reliability of the unlearning process.

**Strengths:**

1. The paper is easy to follow, and the main message is conveyed clearly.

2. The theoretical analysis is rigorous and well-supported by formal proofs.

3. The paper addresses an important and timely problem in the field of machine unlearning.

**Weaknesses:**

1. The main claim—that increasing the amount of public data improves unlearning performance—appears somewhat self-evident. It is conceptually similar to stating that reducing the proportion of forget data helps maintain retain accuracy. This insight has already been discussed in prior theoretical works (e.g., [A], [B]), which diminishes the novelty of the contribution.

[A] Certified Data Removal from Machine Learning Models. (Guo et al.)
[B] Towards Source-Free Machine Unlearning (Ahmed et al.)

2. In Theorem 3.3, the authors show that the expected loss on retain data under the unlearned model is upper bounded by the domain distribution gap between the public and private datasets. However, this result seems somewhat tautological — if the two distributions are identical, it trivially reduces to the case of having less forget data within a unified distribution; if they differ, the bound naturally worsens, which is self-evident. Hence, the practical utility or new insight offered by this theorem is unclear.

3. The paper lacks sufficient experimental validation. Key quantitative results—such as forget accuracy, retain accuracy, and test accuracy across different datasets or classes—are missing. The evaluation is limited to the DomainNet dataset, and no diverse metrics are reported to substantiate the theoretical claims.

**Questions:**

1. In Table 1, the reported numbers appear identical across rows, and the gap values are all 0.0. Could the authors clarify the purpose of this table? If the intention is to show that increasing public data reduces the gap, the results do not reflect that trend. Moreover, as the amount of public data increases, it seems the size of the forget data simultaneously decreases—how is this a fair or controlled comparison?

2. The explanation of the weight distribution mechanism using N models is unclear. A concise pseudocode or algorithmic description would greatly help in understanding this procedure and verifying its theoretical soundness.

---

> ### Author Response · Authors · 2025-11-21
>
> We thank the reviewer for the feedback. Please note our answers and resulting modifications below:
>
> **Weaknesses:**
> - Novelty: we believe the novelty of our work lies not in introducing public data itself, but rather in providing the first formal characterization of how public data impacts the unlearning-utility trade-off through certifiable guarantees.
> The reviewer notes that the intuition—that public data diminishes the "impact" of each data point on the optimized weights—is self-evident. We concur. However, translating this intuition into actionable guarantees requires precisely quantifying how much noise and how many unlearning iterations are necessary in the presence of public data. Prior work by Chien et al. (2024a,b) provided unlearning bounds that remain agnostic to data composition, yielding more pessimistic guarantees. Our Theorems 3.1 and 3.2 demonstrate that these bounds can be tightened by accounting for public data, thereby enabling practitioners to determine concrete hyperparameter choices without unnecessarily amplifying noise.
> Our scope differs from prior work in a subtle way. While the reviewer frames this as analogous to reducing the forget set proportion to maintain accuracy, we study performance specifically on the private data distribution and quantify performance degradation through the explicit dependence on distributional alignment. This distinction matters because it reveals that when public and private data are well-aligned, public data acts as a performance stabilizer. Conversely, when distributions diverge significantly, the penalty becomes transparent, allowing principled decision-making about data composition trade-offs. This provides practitioners with guidance on when asymmetric unlearning is beneficial versus when alternative approaches may be preferable.
>  Guo et al. and Ahmed et al. (2025) both mention public data augmentation only experimentally with pre-trained feature extractors, providing no certifiable guarantees on how public data improves unlearning performance. Their certified results apply only to linear models or frozen representations.
> In contrast, our work provides the first theoretical analysis of public data's role in Langevin unlearning for general non-convex losses. Theorem 3.1 shows that the Rényi divergence between learning and retraining distributions scales quadratically with public data volume, enabling control over unlearning guarantees through data composition rather than noise alone.Theorem 3.3 provides a decomposition of post-unlearning utility into a distribution mismatch penalty and an unlearning approximation error. This reveals precisely when public data improves both efficacy and performance—when distributions align well—and characterizes the trade-off when they diverge. To our knowledge, no prior work has provided such certifiable insights into the interplay between data alignment, unlearning efficacy, and utility preservation.
> - Theorem 3.3' utility: The insight of Theorem 3.3 is its characterization of how post-unlearning performance on *private* data relates to retraining performance on the *training mixture*, when $D∞(Ppriv∥Ppub)D_\infty(P_{\text{priv}} \| P_{\text{pub}})$ remains bounded. This linkage is non-trivial: it allows us to show that even when distributions are moderately misaligned, incorporating public data improves utility after unlearning. When this mismatch is controlled, this gap remains small, implying that the unlearned model performs well on retained private data—without requiring the noise amplification that standard noise-based unlearning would demand.
> We acknowledge that Theorem 3.3 provides a worst-case guarantee that may be loose in practice, particularly under significant distributional shift (as evidenced by Table 1, where empirical performance remains surprisingly consistent despite misaligned domains). We have refined the analysis in the "Retraining performance bound" section to eliminate unnecessary dependence on retraining performance, replacing it with a direct characterization that isolates the distribution mismatch penalty. This allows for clearer separation between the contributions of distributional alignment versus unlearning approximation error.
>
> **Questions**:
> 1- We have addressed the reviewers concerns and provided additional performance metrics and alternative public set proportions in sections 4.2 and 4.3
> 2-  If the concern pertains to Rényi divergence estimation (Section 4.1), the need to train N independent models is directly motivated by our implementation of Birrell et al. (2023)'s variational framework. Since PNGD induces a stochastic distribution over parameters, we obtain samples by independently training models under each regime. We have added pseudocode in Appendix A.7.3.
> If the concern pertains to membership inference attacks (Section 4.3), the multi-model sampling procedure follows standard established practice, as demonstrated in Hayes et al. (2024, 2025) and Carlini et al. (2021).

---

### Official Review · Reviewer_TMUa · 2025-11-01

**Soundness:** 2
**Presentation:** 3
**Contribution:** 2
**Rating:** 4
**Confidence:** 4

**Summary:**

The paper reframes Langevin unlearning under a more realistic asymmetric setting in which a subset of the training data is public and never needs to be forgotten. Theoretical analysis demonstrates that incorporating public data enables better unlearning-utility trade-off without additional noise or restrictive differential privacy assumptions, allowing control over unlearning guarantees through data composition rather than noise amplification. The authors provide a fine-grained analysis of how distributional alignment between public and private data affects this trade-off. Empirical results corroborate the theoretical analysis.

**Strengths:**

1. The paper introduces a realistic asymmetric-unlearning setup that separates public data (never forgotten) from private data (subject to removal), breaking the traditional noise-vs-utility deadlock.
2. The authors provide the first rigorous proof that increasing public-data volume quadratically shrinks the Rényi divergence between the unlearned and retrained distributions.
3. The paper is overall well-organized.

**Weaknesses:**

1. The idea lacks novelty, as the incorporation of public data represents a fairly natural extension and is already a well-established technique in the field of differential privacy. The main part of the theoretical derivation is built upon the analysis of Langevin Unlearning, with only minor modifications to the parameters $m$ and $n$.
2. Section 2.3 provides an insufficiently detailed introduction to Langevin Unlearning. The reviewers suggest supplementing it with a description of LU's training procedure or including pseudocode for the LU algorithm. Additionally, the last paragraph of Section 2.3 contains repetitive content.
3. In Chapter 3, the symbols $\theta_T$, $\mathcal P$, $\pi_0$, and $P_{train}$ are not defined.
4. The reviewer suggests that the citations in Theorem 3.1 and Theorem 3.2 should refer specifically to the relevant theorems in the cited works.
5. The experimental section lacks figures or tables illustrating the unlearning-utility trade-off, as well as results showing how accuracy changes as $ K $ increases.
6. There is a confusing or inconsistent use of $K$ in Table 1.
7. The analysis of Figure 3 should be presented in the main text rather than in the figure caption.
8. The writing in the section “Retraining performance bound” lacks logical coherence with the preceding content.

**Questions:**

1. In Definition 3.1, should $I(Q,P)=E_P[\cdot]$ rather than $E_P[\cdot]$?
2. In Theorem 3.1, is the numerator on the right-hand side of the inequality missing an \(\eta\) term?
3. The paper states: “We evaluate post-unlearning performance on the private data distribution only, reflecting realistic deployment scenarios where the primary concern is maintaining model quality on the sensitive data that remains after unlearning.” The reviewer is confused as to why the focus is solely on maintaining model quality on the sensitive data that remains after unlearning, and not on maintaining model quality on the public data as well, since public data is a part of the training dataset.

---

> ### Author Response · Authors · 2025-11-21
>
> We sincerely thank the reviewers for their thoughtful and constructive feedback. We provide a detailed response to each concern below.
>
> **Novelty:**
>
>
> We appreciate the reviewer's observation regarding the use of existing proof techniques. While public data augmentation is indeed studied in the differential privacy literature, we respectfully note that theoretical explorations of its effect on machine unlearning with certifiable guarantees remain unexplored. We acknowledge that Theorem 3.1, which characterizes the unlearning effect of public data, adapts and corrects  the proof technique of Chien et al. (2024) to our asymmetric setting. We also alleviate the assumption of convergence to a log-Sobolev stationary distribution, assuming that only the weight initialization is LSI. This is true in practice, as Gaussian initialization satisfies LSI.
> However, we would like to emphasize that our primary and novel contribution is Theorem 3.3, which provides the first characterization of how public data affects post-unlearning model utility on private data. This utility-focused analysis, which reveals the fundamental interplay between unlearning efficacy and performance degradation through the distribution mismatch penalty, has not been previously addressed in the unlearning literature. We recognize, as the reviewer correctly notes in point 5 of the weaknesses, that experimental validation of this trade-off remains incomplete. We are currently preparing complementary experiments that will thoroughly validate the practical relevance of Theorem 3.3 and substantiate its central message through empirical evidence.
>
>  **Presentation, Notation, and Technical Corrections**
>
> We have addressed all notation and presentation issues raised:
>
> - Notation Section (2.2): Formally defined all symbols ($\theta_T$, $\mathcal{P}$, $\pi_0$, $P_{\mathrm{train}}$) and clarified their usage throughout the paper.
> - Definition 3.1: Corrected the inequality sign and fixed the typo in the Fisher relative information definition.
> - Theorem 3.1: Added the missing $\eta$ term in the numerator and included specific citations to the relevant theorems in Chien et al. (2024).
> - Theorem 3.2: Similarly updated with precise references to cited theorems.
> - Table 1: Removed the confusing use of the variable $K$ for improved clarity.
> - “Retraining Performance Bound" Section: Expanded with additional context to better motivate the open question we pose: how can the current distribution mismatch bound be linked to a proper generalization bound involving the true risk minimizing distribution.
> -Added pseudo code for the Langevin Unlearning procedure in appendix A.6.
> - Moved the description of Figure 3 to the main body of the experimental section.
>
> **Experiments:**
>
> In response to the reviewer's suggestion that the experimental section lacks sufficient illustration of the unlearning-utility trade-off, we supplement our work in sections 4.2 and 4.3, showing for different data alignment regimes how unlearning and retraining losses relate to each other as $n_{\text{pub}}$ increases. These additions provide empirical evidence supporting the theoretical predictions of Theorem 3.3 and strengthening the practical relevance of our framework. We also address point 7 by moving the analysis of figure 3 to the main text, complementing it with a detailed analysis of the interplay between MIAs and model utility, as the public data distribution shifts from the private data distribution.
>
> **Private Data Performance as the Primary Utility Metric:**
>
>
> The reviewer raises an important question regarding our focus on private data performance. For completeness and transparency, we have provided an equivalent result for the training distribution mixture in Appendix A.4.1. Yet, in practice, private data is inherently scarce and domain-specific—exemplified by medical records and sensitive personal information—while public data is abundant, noisy, and broadly available (e.g., CommonCrawl, ImageNet).
> Evaluating model utility on the training distribution mixture would be misleading in this context. Our framework is designed to leverage the abundance of public data to improve unlearning; as public data volume increases, the training mixture becomes increasingly dominated by public data. A model could perform well on this mixture while degrading precisely on the private data—which is where practitioners require strong performance guarantees.
>
> In practice, deployed models operate on the private data distribution: a medical AI serves patients at hospitals (private distribution), not to classify public benchmark images; a recommendation system personalizes suggestions based on user history (private distribution), not generic public web data. The fundamental utility question practitioners must answer is therefore: Does my model maintain strong performance on my sensitive, high-value private data after unlearning? This is the question our framework directly addresses through Theorem 3.3.

---

### Meta-Review · Area_Chair_M4Q2 · 2025-12-17

**Summary:**

The paper offers several notable strengths. First, it provides a formal study of the unlearning-utility trade‑off when two distinct data sources--public and private--are involved. This asymmetric analysis of Langevin unlearning is novel. In particular, Theorems 3.1 and 3.2 extend (and correct) prior work on Langevin unlearning by incorporating the presence of both public and private data (cf. Chien et al. 2024). Moreover, the new proof relaxes earlier assumptions, requiring the log-Sobolev inequality only at initialization. Finally, the experimental validation has been strengthened, and the paper's presentation benefits from the constructive feedback received during the review process.

Nevertheless, the paper has several notable weaknesses. The theoretical contribution is limited. Very briefly, the $n^2$ factor is replaced by $(n_{priv} + n_{pub})^2$, which is exactly what one would expect, and the theorems merely confirm that larger datasets improve unlearning performance. Crucially, these results do not capture the anticipated advantage of the asymmetric (public vs. private) setting that underlies the proposed approach. The bound presented in Theorem 3.3 becomes non-vacuous only when the distributions of public and private data are nearly identical, suggesting that any observed gains stem primarily from the increased total sample size rather than from exploiting asymmetry. Moreover, because the constants in the bound are intractable to estimate, there is no way to assess empirically how tight the bound is; the experiments indeed reveal that the theoretical guarantees can be extremely loose.
The bounds derived in Koloskova et al. (2025) rely on weaker assumptions. The fact that they do not depend explicitly on the dataset size does not allow us to dismiss the comparison so quickly.

**Reviewer Concerns:**

The authors have answered concerns about the experiments. They have improved the paper a lot.
I've explained what remains unresolved in the summary.

**Reviewer Scores:**

I am inclined to agree with what Reviewer  gRLK has written.: "the paper is more of a strong application paper than a fundamental theory paper.". But the authors have presented their paper as a theoretical one, that's why I am convinced that those who gave a 2 would not have changed their rating.

---

### Decision · Program_Chairs · 2026-01-26

Reject